


# Temporal variation of VOC fluxes measured with PTR-TOF above a boreal forest

Simon Schallhart[1], Pekka Rantala[1], Maija K. Kajos[1], Juho Aalto[2], Ivan Mammarella[1], Taina M. Ruuskanen[1] and Markku Kulmala[1]

[1]Department of Physics, University of Helsinki, Finland
[2]Department of Forest sciences, University of Helsinki, Finland

*Correspondence to*: Simon Schallhart (simon.schallhart@helsinki.fi)

**Abstract.** Between April and June 2013 fluxes of volatile organic compounds (VOCs) were measured in a Scots pine and Norway spruce forest using the eddy covariance (EC) method with a proton transfer reaction time of flight (PTR-TOF) mass spectrometer. The observations were performed above a boreal forest at the SMEAR II site in Southern Finland. We found a total of 25 different compounds with exchange and investigated their seasonal variations from spring to summer. The majority of the net VOC flux was comprised of methanol, monoterpenes, acetone and butene. The VOC emissions followed a seasonal trend, the released amount increased from spring to summer. Only a three compounds were emitted in April while in June emissions of some 19 VOCs were observed. During the measurements in April, the emissions were dominated by butene, while in May and June methanol was the most emitted compound. The main source of methanol is likely the growth of new biomass. During a 21-day period in June, the net VOC flux was 2.3 nmol $m^{-2}$ $s^{-1}$. This is on the lower end of PTR-TOF flux measurements from other ecosystems, which range from 2 to 10 nmol $m^{-2}$ $s^{-1}$. The EC flux results were compared with surface layer profile measurements, an indirect method using a proton transfer reaction quadrupole mass spectrometer, which is permanently installed at the SMEAR II site was used. For most of the compounds the fluxes, measured with the two different methods, agreed well.

## 1. Introduction

Boreal forests, covering approximately $12.2 \cdot 10^6$ $km^2$ of the earth's surface, which is 30% of the world's forest area (Keenan et al., 2015), emit volatile organic compounds (VOCs), which have an important impact on the chemistry and composition of the atmosphere. These emitted compounds can react with $O_3$, $NO_3$ or OH and form a multitude of new VOCs, which can contribute to the formation and growth of aerosol particles and thereby affect climate change (Tunved et al., 2006; Spracklen et al., 2008). An understanding of these emissions and their quantity is necessary for numerical assessments of future climate and air quality (Guenther et al., 2012; Makkonen et al., 2012). The direct measurement of VOC fluxes above the canopy is one key element in this process. Eddy covariance has become a reference method for measuring canopy exchange (Baldocchi, 2014).

The ecosystem scale VOC fluxes result from emissions caused by biological activity and chemistry, which, take place in soil, forest floor and understory vegetation, trees and within the canopy airspace. The prevalent ability to synthesize and release VOCs to the atmosphere varies between species and often changes with seasons and during the lifecycle of a plant (Karl et al., 2003, Hakola et al., 2006, Holzke et al., 2006). The more heterogeneous a habitat, the more potential sources of variation as regards VOC fluxes exist. Although managed boreal forests are commonly considered to be relatively homogeneous habitats, they are still relatively heterogeneous when compared to grasslands or other agricultural areas (Newbold et al., 2015). In addition to the array of tree species, the understory and forest floor vegetation vary significantly



in terms of species selection, luxuriance and coverage as a result of soil properties, orientation, canopy coverage and water availability. Even a single species, as exemplified for Scots pine, may express considerable intra-species variation in terms of emission composition (Bäck et al., 2012) and capacity (Aalto et al., 2014, 2015). Soils and forest floors are very complex but as yet poorly understood sources of VOCs (Aaltonen et al., 2011, 2013); however, the diverse forms of

microbial activity seems to play a central role in VOC emissions from the soil (Bäck et al., 2010). In this context, above canopy VOC flux measurements represent a vast array of signals from biogenic activity, which largely complicates drawing conclusions on the effects of biological and ecophysiological phenomena on VOC emissions from forests. However, the above canopy VOC flux method is a crucial tool in studying VOC emission in a continuum from the sub-cellular scale up to the effects of VOCs on atmospheric composition and processes on local, regional and global scales.

VOC concentrations and emissions have been measured during campaigns since 1998 at the SMEAR II (Station for Measuring Ecosystem-Atmosphere Relations). The VOC flux measurements at the station have consisted of different instruments and techniques: disjunct eddy covariance (DEC; e.g. Rinne et al., 2007), the surface layer profile (SLP) method (e.g. Rantala et al., 2014) using a Proton Transfer Reaction-Quadrupole (PTR-Quad) mass spectrometry and a gas chromatograph mass spectrometry using the gradient method (e.g. Rinne et al., 2000). In recent years, proton transfer

reaction-time of flight (PTR-TOF) mass spectrometers have been used to measure eddy covariance fluxes (Ruuskanen et al., 2011; Kaser et al., 2013a; Park et al., 2013; Brilli et al., 2015; Schallhart et al., 2016), but the number of these studies is still low. With this new measurement setup, it is possible to identify the elemental composition of the compounds with detectable fluxes. Furthermore, the preselection of the measured compounds is no longer necessary, as the PTR-TOF measures full mass spectra. Its ability to measure with 10 Hz time resolution leads to noise reduction compared to the

DEC method, which uses a lower measurement frequency (typically 0.05 to 1 Hz; e.g. Rinne and Amman, 2012).

This study used a PTR-TOF to measure VOC fluxes during April (9 days), May (9 days) and June (21 days) 2013 and these are the first results from EC measurements of VOCs above a boreal forest. The main objective was to investigate how the fluxes and compounds with flux behavior change during the transition from winter to summer. A second objective was to compare these PTR-TOF EC measurements with the long-term PTR-Quad measurements of SLP fluxes. Finally,

the results from the 21 days of flux measurements in June were compared with other VOC flux studies using EC and PTR-TOF.

## 2. Methods

### 2.1. Site description

The measurements were carried out from April until the end of June 2013 at the SMEAR II (Station for Measuring

Ecosystem-Atmosphere Relations) in Hyytiälä, southern Finland. The station is 180 m above sea level and located in the middle of a *Pinus sylvestris* (Scots pine) dominated stand, while *Picea abies* (Norway spruce) covers 15% of the forest. In addition to the dominating Scots pine and Norway spruce, other tree species are present at the study site e.g. *Betula pendula* and *Betula pubescens* (silver and downy birch), *Populus tremula* (trembling aspen), *Sorbus aucuparia* (rowan) and *Salix caprea* (goat willow). The typical tree height is about 18 m and the surroundings have been described by Hari

and Kulmala (2005) and Ilvesniemi et al. (2010). The climatological mean annual temperature is 3.5°C and the mean annual precipitation is 711 mm (Pirinen et al., 2012). The measurements were conducted on a scaffold tower (61.847407°N, 24.295150°E) and the inlet height was 23 m. The temperature varied between -2°C and 27°C during the measurement periods and the main wind direction was south-southwest.



## 2.2 Flux measurement setup

The measurements were conducted with a PTR-TOF 8000 (Ionicon Analytic GmbH; Jordan et al., 2009; Graus et al., 2010). It was operated with a drift tube voltage of 600V and a drift tube pressure of 2.3 mbar. Together with the drift tube temperature of 60°C, the $E_{PTR}/N_{PTR}$ ($E_{PTR}$ being the electrical field strength and $N_{PTR}$ the gas number density) was calculated to be 130 Td. The instrument was placed in an air-conditioned cottage next to the measurement tower. Sample air was pumped through a 30 m long (8 mm inner diameter, i.d.) PTFE inlet, with a flow of 20 L min$^{-1}$. To prevent condensation on the inlet walls, the tube was heated with an 8 W/m passive heating wire. A subsample of 0.5 L min$^{-1}$ was collected via a 10 cm PTFE tubing (1.6 mm i.d.), which lead over a three-way valve (type 6606 with ETFE, Bürkert GmbH & Co.KG) and 20 cm of PEEK tubing (1 mm i.d.) to the PTR-TOF. The PTR-TOF data was analyzed with the tofTools, which are described in more detail in Junninen et al., 2010. A 3-d ultrasonic anemometer (HS-1199, Gill instruments) was placed 10 cm above the inlet. Both VOC and wind measurements were recorded at 10 Hz resolution. In addition, eddy covariance fluxes of carbon dioxide ($CO_2$) are routinely measured at the site using a closed-path infrared gas analyzer (Licor 6262, USA; Mammarella et al., 2009). $CO_2$ fluxes were calculated by using EddyUH software (Mammarella et al. 2016).

The instrumental background of the PTR-TOF was measured by guiding ambient air through a catalytic converter, which removes the VOCs. This VOC free air was measured three times a day, starting at 00:02, 08:02 and 17:02 and each measurement session lasted for 25 min. This led to a reduced amount of flux data during these hours. Switching between ambient air and the VOC-free air was done with a 3-way valve (type: 6606 with ETFE, Bürkert GmbH & Co. KG) controlled by the PTR-Manager (Ionicon Analytic GmbH, Austria).

For calibration this VOC free air was mixed with a calibration gas (Apel Riemer Environmental Inc., USA) containing 16 different compounds. For uncalibrated compounds the sensitivities were categorized into three groups $C_xH_y$ (calculated from isoprene, benzene, toluene, o-xylene, trimethylbenzene, naphthalene, α-pinene combined with $C_6H_9$ fragment), $C_xH_yO_z$ (based on acetaldehyde, acrolein, acetone, 2-butanone) and $C_xH_yN_z$ (set to acetonitrile) similar to the setup Schallhart et al. (2016). The average sensitivities for the different compound groups were: $11.4 \pm 2.5$ ncps/ppb for $C_xH_y$, $18.6 \pm 3.1$ ncps/ppb for $C_xH_yO_z$ and $17.7 \pm 2.0$ ncps/ppb for $C_xH_yN_z$. Overall the sensitivities are comparable with Schallhart et al. (2016), only the standard deviation increased, due to the longer time period of the measurements. The setup for background and calibration measurements is described in more detail in Schallhart et al. (2016).

The PTR-TOF is unable to identify the structure of measured compounds, therefore the identification of specific compounds is not possible. In the remaining manuscript, mass 69.0699 amu with the elemental composition $C_5H_9^+$ is called isoprene, even though 2-methyl-3-buten-2-ol (MBO) fragments to the same mass and has a substantial influence on the signal (e.g. Kaser et al., 2013b). Similarly, the mass 93.0699 amu with the elemental composition $C_7H_9^+$ is called toluene, even though p-cymene fragments may affect the signal (Tani et al., 2003). Formaldehyde has only a slightly higher proton affinity compared with the primary ion and therefore back reactions, which are humidity dependent, from protonated formaldehyde to water occur (de Gouw and Warneke, 2006; Inomata et al., 2008; Vlasenko et al., 2010). This may lead to an artificial flux, which is caused by water vapor fluctuations. Therefore, the formaldehyde fluxes are very uncertain. A possible compound for mass 57.0699 with a protonated composition of $C_4H_9^+$ is butene, but it also could be a fragment of butanol (see Sect. 3.4). The monoterpenes ($C_{10}H_{17}^+$) were measured for mass 137.1325 only. The fragments for mass 81.0699 ($C_6H_9^+$) were identified by their Pearson correlation (30 min integrated data) with the signal at 137.1325 of 0.99 and were disregarded from further analysis.



**2.3 Flux and lag time calculations**

VOC fluxes were derived using the eddy covariance (EC) method (e.g. Aubinet et al., 2012). The EC flux is calculated using the covariance:

$$\overline{w'c'}(\lambda) = \frac{1}{n}\sum_{i=1}^{n} w'(i - \lambda/\Delta t)c'(i),\qquad(1)$$

where $w'$ and $c'$ are high frequency fluctuations of vertical wind and concentration, respectively, $i$ the number of the measurement, $n$ the sum of all measurements during the flux averaging time (30 min in this study; $n$ = 18 000), $\Delta t$ the sampling interval (0.1 s) and $\lambda$ the lag time caused by the sampling system. The cross covariance functions (CCF) were calculated by varying $\lambda$ from -200 s to 200 s. In this study, vertical wind and VOC concentrations were both recorded at 10 Hz frequency. The flux calculation procedure is called the automated method and is similar to that in Park et al. (2013) and Schallhart et al. (2016), only the lag time was determined differently.

The VOC and the wind data were recorded on two different computers and their clocks shifted considerably (continuous shift of 2 to 5 s day$^{-1}$; Fig. 1a). Finding the correct lag time is especially challenging when the flux is close to the detection limit. To estimate the proper lag times, three corrections were made:

I.  First, the artificial clock shifting was removed using linear regressions. Therefore, the regressions from the monoterpene CCFs were used to correct the CCFs from all compounds (Fig. 1b).

II. In the second step, an average, absolute CCF was calculated (Fig. 2) for each compound. For this the absolute value of each 30 min CCF between 10:00 and 16:00 was taken and then all the absolute CCFs for the time period of interest were averaged. To reduce the influence of noise, especially when small fluxes are measured, a running mean (±24 step averaging) was used. Then the position of the maximum was searched for a ±10 s time window and used as the lag time. This lag time was calculated for each month and each compound separately.

III. The third and last step was used to correct for smaller shifts, in case the first correction, with the linear regression, was not precise enough. Therefore, each individual 30 min CCF was smoothed by a running mean (±24 step averaging) and the location of the maximum in a ±10 s time-window around the previously calculated lag time was recorded (Taipale et al., 2010) as shown for the monoterpenes in Fig. 1c.

To classify how many of the hundreds of measured compounds show an exchange, a method described in Park et al., (2013) and Schallhart et al., (2016) was used. This method compares the maximum of the averaged, absolute CCF with a certain noise threshold. To reduce the impact of noise, the average, absolute CCF was smoothed (±12 step averaging) and the location of the maximum in a ±10 s time window detected. This position was used in the average, absolute CCF (not smoothed) and compared with the $\sigma_{noise}$ (standard deviation of the noise). The $\sigma_{noise}$ was calculated for 60 s at the borders of the average absolute CCF. If the ratio between the calculated maximum and the $\sigma_{noise}$ was higher than three, the compound was classified to have detectable flux (Fig. 2). This method was applied to flux determinations for each month separately.

The flux underestimation caused by high frequency attenuation was estimated using a parametrization described by Horst (1997). The method uses a system response time and information about stability and horizontal wind speed for estimating the attenuation. The system response time was determined to be around 1.2 s for monoterpenes and the same response time was also used for all the other compounds. However, the response time of e.g. water soluble compounds might be larger due to desorption and absorption processes on the tube walls, as a function of relative humidity and sampling line aging (Mammarella et al, 2009; Nordbo et al., 2013 and 2014). This potentially causes errors of a few percentage on the flux values. However, we expect the effect to be reduced, because a heated sampling tube was used. The low-frequency





corrections that are based on theoretical transfer function shapes (e.g. Rannik and Vesala, 1999), were not applied in this study.

One should note that the determined response time describes the flux attenuation of the whole measurement setup, including the tubing, a horizontal separation between the inlet and the anemometer and the instrument itself. Thus, the response time of this study cannot be used in the case of other PTR-TOF measurements at other locations. In this case, the average attenuation factor was 18%. On average, the correction factor was smaller during the day and larger at night.

**2.4 Flux quality criteria**

The measured fluxes were filtered by three quality criteria, to reduce the systematic uncertainty and ensure their representativeness:

First, the data were flagged if the tilt angle, resulting from the coordinate rotation of sonic anemometer wind velocity components (Kaimal and Finnigan, 1994), was more than 5°, which was the case for 11.9% of the data. Second, all 30 min records with a friction velocity less than 0.2 m s$^{-1}$ were flagged. Following this, 11.2% of the data was flagged. Finally, the flux steady-state test was applied according to Foken and Wichura (1996). All flagged flux values were removed from further analysis. The rejection rate between April and June was 34.1%, 35.1% and 30.5% for acetone, butene and the monoterpenes, respectively. For the monoterpenes, the rejection rates in April, May and June were 19.1%, 17.6% and 30% (daytime) and 25.6%, 24.0% and 43.7% (nighttime).

**2.5 Flux selection**

VOC emissions from April, May and June 2013 were selected for a monthly comparison. Because of technical problems with the anemometer in May, only a nine-day period, from 04 May 2013 to 24 May 2013, could be used. The standard deviation of noise in the averaged, absolute CCFs ($\sigma_{noise}$) determines the exchange threshold and is directly dependent on the amount of data. Therefore, the same amount of data (423 × 30min files) was selected to represent each month and make the comparison with the other months possible. For all three months, the absolute mean of the CCFs between 10:00 and 16:00 (UTC+2) was used to find compounds with statistically significant flux (Park et al., 2013; Schallhart et al., 2016). For April, the measurements were from 14 April 2013 to 24 April 2013 and for June the hottest period was selected, from 01 June 2013 to 12 June 2013.

**3. Results and Discussion**

**3.1 VOC flux variation during the campaign**

Overall 22 compounds with flux were found, of which 16 were identified by their elemental compositions (Table 1). Five compounds, $C_1H_3O_1^+$ (formaldehyde), $C_1H_4O_1N_1^+$ (formamide), $C_4H_7O_1^+$ (crotonaldehyde) and two unidentified peaks with the masses of 84.95 amu and 118.9456 amu, had a negative net flux, each contributed around 1% or less to the total net flux (Fig. 3). As expected the average net flux increased from April (0.66 nmol m$^{-2}$ s$^{-1}$) to May (1.22 nmol m$^{-2}$ s$^{-1}$) and June (3.00 nmol m$^{-2}$ s$^{-1}$). The compounds with detectable flux increased from three in April to 12 in May and 19 in June. Over 80% of the net flux comprised of methanol, acetone, monoterpenes and butene. Of those four main compounds, acetone and monoterpenes had similar emission patterns (based on the total net flux) over the measurement period, while methanol had no detectable flux in April. The development of the diurnal cycle can also be seen in Fig. 4, as April had a minor flux variation between day and night, whereas May and June showed a clear dependence on the temperature. The





maximum emission was detected between 14:00 and 16:00 (Fig. 4); this is in agreement with the fact that VOC synthesis is driven by temperature and light (Ghirardo et al., 2010; Taipale et al., 2011), while potential evaporation from storage pools is primarily driven by temperature alone (Guenther et al., 1993). The maximum temperatures were typically measured in mid-afternoon, when the light availability has still not yet decreased to a great extent when compared to the

light conditions around noon. This can be seen in Fig. 5 where the highest emissions of the globally most emitted VOCs (excluding methane), the group of monoterpenes and isoprene, are during the hottest period of the campaign. Furthermore, the high monoterpene emissions during low PAR (<200 µmol m$^{-2}$ s$^{-1}$; grey data points in Fig. 5) conditions can be explained by pool emissions, whereas the de novo isoprene emissions during this time were low. Unlike the maximum emission time, that was similar for all month, the minimum net flux was between 20:00 and 21:00 in April, 03:00 and

04:00 in May and 01:00 and 02:00 in June. Table 1 shows all the compounds with detectable flux for the three months and their 24h average emission and deposition.

### 3.1.1 Low emissions during snow melt

As expected the total emission (0.68 nmol m$^{-2}$ s$^{-1}$) was smallest in April (compared to May and June). The snow melted during this period and the average temperature and photosynthetically active radiation (PAR) were at their lowest, 4.4°C

and 268 µmol m$^{-2}$ s$^{-1}$, respectively. The total deposition (-0.01 nmol m$^{-2}$ s$^{-1}$) was also weakest. Butene (C$_4$H$_9^+$) dominated the emissions, comprising over 60% of the net flux. C$_4$H$_9^+$ had the highest emissions between 14:00 and 15:00 with 0.82 nmol m$^{-2}$ s$^{-1}$ and the lowest between 21:00 and 22:00 with 0.08 nmol m$^{-2}$ s$^{-1}$. Acetone contributed almost 20% to the emissions and was the only compound in April for which a diurnal deposition was detected. Between 22:00 and 23:00 the measured flux reached a minimum of -0.12 nmol m$^{-2}$ s$^{-1}$, whereas between 13:00 and 14:00 the emission peaked with

0.40 nmol m$^{-2}$ s$^{-1}$. The heaviest measured compounds with detectable flux was the group of monoterpenes, which contributed 17% to the total emission and had the highest emissions between 14:00 and 15:00 with 0.24 nmol m$^{-2}$ s$^{-1}$. The lowest emissions of 0.06 nmol m$^{-2}$ s$^{-1}$ were measured during morning between 06:00 and 07:00.

### 3.1.2 Increase of emissions at start of growing season

Methanol is released in plant growth (e.g. Galbally and Kirstine, 2002), its emissions increased from being not detectable

in April to a late afternoon maximum of 2.11 nmol m$^{-2}$ s$^{-1}$ in May. Compared to April's emissions the order of the major emitters reverses in addition to the increase of compounds with observable exchange. In May night temperatures were above zero, while the sun warmed late afternoons to around 20°C. The mean temperature was 11.4°C and the mean PAR was 301 µmol m$^{-2}$ s$^{-1}$, during the measurements in May. The total deposition was higher than in the other months and reached -0.13 nmol m$^{-2}$ s$^{-1}$, which is less than 10% of the total emission (1.37 nmol m$^{-2}$ s$^{-1}$). Methanol was the most

emitted (35%) and deposited compound (49%) in May, with the highest deposition of -0.47 nmol m$^{-2}$ s$^{-1}$ between 11:00 and 12:00. This deposition can be explained by the occurrence of rain during or right before this time window, which happened twice during the 9-day measurement period. The water-soluble methanol is suspected to be dry deposited on the wet surfaces in the forest, (Laffineur et al., 2012; Wohlfahrt et al., 2015; Schallhart et al., 2016). The monoterpenes were the second most emitted compound group and contributed 23% to the total emission. Their maximum emission was

0.76 nmol m$^{-2}$ s$^{-1}$ between 15:00 and 16:00 and the minimum emission was 0.13 nmol m$^{-2}$ s$^{-1}$ between 3:00 and 4:00. Recently, Aalto et al. (2014, 2015) have shown that Scots pine needles are a pronounced source of monoterpenes in spring already before growth onset, and especially after bud break, when the formation of new biomass releases large amounts of terpenoids and other VOCs. The results of this study are in general consistent with those findings in terms of detected



mean fluxes and diurnal patterns. During the start of the growing season, acetone was the third most emitted compound. In May it had the maximum emission of 0.61 nmol m$^{-2}$ s$^{-1}$ between 10:00 and 11:00 and the minimum between 03:00 and 04:00 with 0.04 nmol m$^{-2}$ s$^{-1}$. The flux of C$_4$H$_9^+$ decreased by almost two-thirds compared to April comprising 11% of the total emission. The maximum flux of 0.29 nmol m$^{-2}$ s$^{-1}$ was between 15:00 and 16:00 and the minimum of 0.02 nmol

m$^{-2}$ s$^{-1}$ between 22:00 and 23:00. Formamide passed the 3 $\sigma_{noise}$ criteria only in May, where it explained 2% of the total emission and 24% of the total deposition. The emissions were highest between 19:00 and 20:00 with 0.13 nmol m$^{-2}$ s$^{-1}$ and the deposition peaked between 12:00 and 13:00 with -0.20 nmol m$^{-2}$ s$^{-1}$. Some of the emissions, e.g. butene which is discussed in Sect. 3.4, were not related to the start of the growing season. However, most of the emissions, were biogenic.

### 3.1.3 Maximum emissions during summer

During the growing season in the first weeks of June, the highest total emission, of the campaign, 3.12 nmol m$^{-2}$ s$^{-1}$, was recorded. The average temperature and PAR were highest in this period being 17.2°C and 466 µmol m$^{-2}$ s$^{-1}$, respectively. Temperature and light are the drivers of biogenic emissions (Guenther et al., 2012), so as expected the diurnal maximum of the net flux, 9.69 nmol m$^{-2}$ s$^{-1}$, was observed on the afternoon in June. The four most emitted compounds had the same order as in May and all the emissions of June's ten most emitted compounds increased when compared to May, even if

their relative (to the net flux) emissions (Table 1) decreased. In the growing season, methanol was the most emitted compound, comprising 39% of the total emission and 17% of the deposition. The methanol flux was highest between 15:00 and 16:00 with 4.56 nmol m$^{-2}$ s$^{-1}$, while between 03:00 and 04:00 it was deposited (-0.26 nmol m$^{-2}$ s$^{-1}$). Growing leaf biomass is expected to release methanol due to cell wall demethylation (Galbally and Kirstine, 2002, Hüve et al., 2007, Aalto et al., 2014). The increase of methanol fluxes from undetectable in April to about 0.5 nmol m$^{-2}$ s$^{-1}$ in May

and finally well above 1 nmol m$^{-2}$ s$^{-1}$ in June coincides with the typical coniferous needle biomass growth onset at the beginning of May and maximum needle elongation rate around mid-summer (Aalto et al., 2014).

Other biogenic emissions include the group of monoterpenes contributing 20% and acetone contributing 13% to the total emission. Similar to methanol, the highest monoterpenes and acetone emissions were observed in the late afternoon, with 1.46 nmol m$^{-2}$ s$^{-1}$ and 1.08 nmol m$^{-2}$ s$^{-1}$, respectively. C$_4$H$_9^+$ was the fourth most emitted compound with 8.2% of the total

emission and a flux between 0.70 nmol m$^{-2}$ s$^{-1}$ and 0.02 nmol m$^{-2}$ s$^{-1}$. Formaldehyde contributed over 40% to the total deposition in June, resulting in a flux minima of -0.23 nmol m$^{-2}$ s$^{-1}$. However, formaldehyde flux measurements are uncertain as discussed in Sect. 2.2.

When we compared the selected 9 days to all measurements in June (21 days; Fig. 6), we found that five compounds no longer fulfilled the 3 $\sigma_{noise}$ criteria. Using the longer period lead to the rejection of formaldehyde, phenol, p-cymene and

two unidentified masses 84.9500 amu and 99.0201 amu (Table 1 and Table 2). The rejection of formaldehyde was the major reason for the change of deposition from -0.13 nmol m$^{-2}$ s$^{-1}$ to -0.09 nmol m$^{-2}$ s$^{-1}$. Three new masses had a detectable flux, these were acetonitrile (42.0338 amu) and two unidentified masses 89.0386 amu and 99.0769 amu. The total emission decreased from 3.12 nmol m$^{-2}$ s$^{-1}$ to 2.36 nmol m$^{-2}$ s$^{-1}$. This difference can be explained by the lower average temperature of 15°C and the lower PAR of 406 µmol m$^{-2}$ s$^{-1}$ during the longer period.

### 3.2 Eddy covariance VOC fluxes above different ecosystems

Ecosystems and their phenomenology define which volatile organic compounds are released. In this study 25 compounds were exchanged, with 17 of them emitted during June. The measured emissions and the observed amount depend on many environmental aspects as well as meteorology, experimental setup (e.g. inlet length) and length and time of the





measurements. The 24h net emission during 21 days in June were 2.27 nmol m$^{-2}$ s$^{-1}$, which is on the low side compared to other PTR-TOF fluxes from other ecosystems (Table 2). In the *Pinus sylvestris* (Scots pine) forest in Hyytiälä, the major emissions in June were from methanol, monoterpenes, acetone and butene. Measurement gaps were excluded, when calculating the length of the data sets. Most of the studies used a data set between 20 and 35 days, Brilli et al. (2015) being an exception with 129 days. All measurements were carried out around summer, when the plant activities were high. This study was using data from June, Park et al. (2013) and Schallhart et al. (2016) used data from June and July, Kaser et al. (2013a) data from August and September and Brilli et al. (2015) used data from June until the end of October. Out of these studies, the lowest 24h net flux was measured at a 2-year-old *Populus* (poplar) plantation (Brilli et al., 2015) in Belgium, with 1.99 nmol m$^{-2}$ s$^{-1}$. Isoprene was emitted most, followed by methanol, acetone and the group of green leaf volatiles (measured via a fragment). The low emission can be partly explained by the long measurement period that extended over the summer. Park et al. (2013) reported a net flux of 4.43 nmol m$^{-2}$ s$^{-1}$ above an orange grove. The most emitted compounds were methanol, acetic acid, monoterpenes and acetone. The study was exceptional, as it measured significant fluxes for several hundreds of VOCs. Despite that, the highest net flux was measured by Schallhart et al. (2016) above a mixed *Quercus* (oak) *Carpinus betulus* (hornbeam) forest with 9.78 nmol m$^{-2}$ s$^{-1}$. In their study the most emitted compounds were isoprene, methanol, acetone and methyl vinyl ketone+methacrolein. The high emissions can be explained by the ecosystem, as oaks are known to be strong isoprene emitters (e.g. Potosnak et al., 2014). Kaser et al. (2013a) reported 8h daytime fluxes only, so a direct comparison with the 24h net flux is not possible. However, the *Pinus ponderosa* (Ponderosa pine) flux was dominated by MBO+isoprene fluxes, followed by methanol and acetic acid.

The net carbon flux of the VOCs during the campaign in Hyytiälä was 8.04 nmol C m$^{-2}$ s$^{-1}$ (Fig. 7). The group of monoterpenes was the highest emitter of carbon with 54% of the net carbon exchange, followed by methanol (12%), acetone (11%), butene (10%) and isoprene (5%). Acetaldehyde, the $C_3H_5^+$ fragment and toluene contributed 2% respectively, while acetic acid and the sum of the remaining compounds both contributed 1%. Compared to the $CO_2$ net ecosystem exchange (NEE) of -4266 nmol C m$^{-2}$ s$^{-1}$, the carbon released as VOC represents less than 0.2% of the NEE of the corresponding period. In Brilli et al. (2015) the VOCs, with 6.36 nmol C m$^{-2}$ s$^{-1}$, represent 0.8% of the carbon exchange and in Schallhart et al. (2016) VOCs had a carbon flux of 41.8 nmol C m$^{-2}$ s$^{-1}$, which corresponded to 1.7% of the NEE. There was an order of magnitude difference between the proportion of net assimilated carbon released as VOCs between a Mediterranean oak-hornbeam forest (Schallhart et al., 2016) and a boreal evergreen forest. Also, a middle European poplar plantation (Brilli et al., 2015) clearly released a higher proportion of the assimilated carbon as VOCs than a boreal site in this study. These findings strongly imply that there are significant differences between the ecosystems in how they allocate carbon to VOCs, however the reasons for that are rather related to light and thermal conditions, species selection and age structure and soil properties than to the efficiency of an ecosystem to produce VOCs as an indefinable concept. In boreal ecosystems with relatively northern locations, the majority of carbon assimilation is concentrated within a couple of months around mid-summer with very short nights. The structure of forest and the tree species are effective in utilizing the high light availability during the summer months, whereas in more southern locations the light availability and thermal conditions allow more even carbon assimilation throughout a considerably longer period. This partly explains the lower proportion of C released as VOCs determined in this study, when compared to those measured in more southern locations. Additionally, the constitutive emission capacities of boreal evergreen species are known to be low when compared to deciduous species more common in central and southern Europe (Rinne et al., 2009; Ghirardo et al., 2010).





### 3.3 Comparison between PTR-TOF and PTR-Quad measurements

Generally, the EC method detected more masses than the SLP measurements, which can be explained by the preselection of the masses to be measured by the PTR-Quad and its low duty cycle. The comparison between PTR-TOF using the EC method and the PTR-Quad using surface layer profile method (SLP, see Rantala et al., 2015) between April and June revealed 12 more compounds with exchange (see Table 1, 2 and 3). The PTR-TOF measures all VOCs in a certain mass range, while the PTR-Quad needs a preselection of masses, limiting the number of recorded compounds. Even though the EC method found almost twice the number of compounds with exchange, the total exchange was in the same order of magnitude for both methods. The discrepancy of the results of the two measurements are mainly due to instrument and method differences. The horizontal distance between the inlets of these two instruments was just 25 m and the PTR-Quad was measuring from 13 m higher (calculated height 36 m) than the PTR-TOF, leading to a larger footprint area for the SLP method.

The main compounds (methanol, acetone and monoterpenes) were detected with both methods. $C_4H_9^+$ (57.0699 amu) could not be measured by the PTR-Quad, as the unit mass resolution of the instrument is unable to separate the signal from the water cluster isotope $H_7O_2^{18}O_1^+$ (57.0432 amu). On the other hand, the SLP measurements observed a flux of ethanol+formic acid which was not detected by the PTR-TOF. Interestingly, formic acid fluxes have also been observed at SMEAR II using the EC method with an Iodide-Adduct High-Resolution Time-of-Flight Chemical Ionization Mass Spectrometer in 2014 (Schobesberger et al., 2016). Other VOC fluxes, which were not detected by the PTR-TOF were methyl ethyl ketone (73 amu), a fragment of the green leave volatiles (83 amu) and MBO (87 amu). The formaldehyde fluxes were not included in the comparison, as the PTR-TOF detected them just during the first 8 days in June, during which the PTR-Quad had technical problems resulting in less than ten overlapping data points from the two instruments. The magnitude of the studied fluxes as well as their diurnal patterns were comparable for methanol, acetone, isoprene and monoterpenes (Table 3 and Fig. 8). The monoterpene and methanol fluxes had also a good correlation with each other (Fig. 9).

The fluxes of acetonitrile, acetaldehyde, acetic acid and toluene did not agree similarly well. Unfortunately, the SLP measurements were not working during a warm, i.e. high flux, period in the beginning of June. Thus, the comparison was done using flux values that were mostly close to the detection limits of the EC and the SLP setups. Therefore, the correlations between the methods were poor for rest of the compounds. The toluene flux discrepancy was likely caused by a different detection of the toluene signal in the two instrument and not due to the different flux methods. The toluene measurements at mass 93 with PTR-Quad and 93.0454 with PTR-TOF resulted in different concentration values and should be handled with care. Different fragmentation from higher masses (most probably p-cymene), influence from two other mass peaks seen at nominal mass 93 (92.5 amu to 93.5 amu) and/or unsuccessful calibrations may probably explain the observed differences. Kajos et al. (2015) reported similar discrepancies in toluene concentration measurements with the PTR-Quad at the site. Also acetic acid is fragmenting when measured with the PTR method (Baasandorj et al., 2015). Higher fragmentation in the PTR-TOF (61.0284 amu), when compared to the PTR-Quad (60.5 amu to 61.5 amu), could account for a part of the lower acetic acid fluxes with the EC method. However, even if the acetic acid main fragment $C_2H_3O_1^+$ (Baasandorj et al., 2015) is taken into account, in addition to the signal from the parent mass, the EC fluxes are still lower than the SLP results. Another uncertainty comes from the lack of acetic acid in the calibration standard. For the uncalibrated compounds the sensitivity can be calculated (Sect 2.2; Rantala et al., 2015), they, however, are more uncertain and can lead to systematic discrepancies. A recent study compared PTR-TOF measurements with gas chromatography mass spectrometer measurements and also in this study the PTR-TOF underestimated the acetic acid





concentration (Helén et al., 2016). The authors suggest that a possible memory effect in the inlet or instrument could lead to this underestimation, which was also reported by de Gouw et al. (2003). This effect could lead to an additional attenuation of the acetic acid signal, and thus decrease the measured flux.

In addition to the differences between the PTR-Quad used in SLP and the PTR-TOF used in EC, the flux calculation methods could also lead to discrepancies: i) the SLP fluxes have larger footprints, which was seen as different flux values, ii) either the SLP or the EC method work improperly for these compounds. One should also note that the comparison was based on a small data set only (Table 3) and, thus, random variations also affect the results. The net flux for all compounds in Table 3 was 1.120 nmol m$^{-2}$ s$^{-1}$ for the PTR-TOF and 1.471 nmol m$^{-2}$ s$^{-1}$ for the PTR-Quad. If the net flux is calculated for all the compounds, which were measured by the individual instrument, it is 1.476 nmol m$^{-2}$ s$^{-1}$ for the PTR-TOF and 1.802 nmol m$^{-2}$ s$^{-1}$ for the PTR-Quad.

### 3.4 Anthropogenic flux of $C_4H_9^+$

The identification of the compound with the elemental composition of $C_4H_9^+$ is problematic as it could be protonated butene, which can be emitted by forests (Goldstein et al., 1996; Hakola et al., 1998) and from anthropogenic sources (Harley et al., 1992; Na et al., 2004). Another possible contribution to the $C_4H_9^+$ signal comes from the fragmentation of butanol, which as many other alcohols, can lose an OH during ionization (Spanel and Smith, 1997). Denzer et al. (2014) reported that the most abundant signal of butanol, when measured with PTR-Quad, is for the $C_4H_9^+$ mass. Fragmentation tests using the PTR-TOF confirmed this. In Fig. 10 the average $C_4H_9^+$ flux from different wind sectors and the windrose for the individual 30 min $C_4H_9^+$ fluxes are shown. The sources of the $C_4H_9^+$ clearly lay in the western part of the forest as there are just low emissions and depositions in the east side. The highest average flux of $C_4H_9^+$ came from south-southwest (195°) with an average flux of 0.57 nmol m$^{-2}$ s$^{-1}$ while another maximum lay in north-northwest (345°) with 0.42 nmol m$^{-2}$ s$^{-1}$. The cottages where the butanol using aerosol measurements, condensation particle counters (CPCs), of the station are located (Fig. 10, orange circles), lie approximately in these directions. An additional CPC was mounted on a mast located west of the VOC flux measurements. Therefore we conclude that the $C_4H_9^+$ signal is mainly from butanol used by the aerosol instruments and thus the flux is anthropogenic. During 2013, approximately 100 L of butanol were evaporated in the CPCs at the station. During our measurements the contribution of the butanol fragment to the total emission was 63% in April, 11% in May and 8% in June.

### 4. Conclusion

Overall, the exchange of 25 compounds was observed over a boreal Scots pine forest. During the transition from early spring to mid-summer the net flux increased by a factor of five and the number of compounds changed from three to 19. The highest emissions occurred in late afternoon, while deposition was observed mainly at night. The majority of the net VOC flux was comprised of methanol, monoterpenes, acetone and butene. The measured butene flux was most likely a fragment of butanol and created by evaporation in the particle counters used at the SMEAR II station. Twelve compounds were measured either only in May or June, which implies a strong seasonal cycle and a high diversity of VOC emissions from the boreal forest in Hyytiälä.

Compared to EC fluxes from other ecosystems measured with the PTR-TOF, the VOC emission in the boreal forest was small, 2.36 nmol m$^{-2}$ s$^{-1}$, even though the measurements in June had the longest day length, up to 19.5 h. In relation to the $CO_2$ exchange, the VOCs are only less than 0.2% compared to the net ecosystem carbon exchange.



The EC fluxes measured with PTR-TOF and the SLP fluxes measured with PTR-Quad had similar results for the main flux compounds: methanol, monoterpenes and acetone, thus, confirming the feasibility of the indirect SLP method at the site. For small fluxes, like acetonitrile, isoprene and acetaldehyde the results are affected by noise. Toluene and acetic acid show significant differences, which could hint at differences in the fragmentation patterns of the instruments. Further

5    research is still needed to close the gap between the fluxes measured by the two instruments.

Therefore, longtime measurements with the PTR-Quad or other instruments, which create less data and do not need such work intensive data post processing as the PTR-TOF, are essential. If a research network of sites with VOC flux measurements is established in the future, cheaper and easier to use instruments are needed. Still, intensive campaigns with more selective instruments are an important asset to understand biosphere-atmosphere exchanges and air chemistries

10    in different ecosystems.

**Acknowledgements**

We would like to thank Heikki Junninen and the tofTools team for providing tools for mass spectrometry analysis. We are further grateful to the Hyytiälä staff, especially to Janne Levula, Heikki Laakso, Matti Loponen and Reijo Pilkottu for all the help at the SMEAR station, as well as Pasi Aalto, Erkki Siivola and Frans Korhonen for their technical help in

15    Helsinki. Also thanks to Pasi Kolari and Petri Keronen for providing the meteorological data. This research received funding from the Academy of Finland Centre of Excellence program (project number 272041).



**Table 1: The exchange of the different compounds in April, May and June. All the presented emission (E) and deposition (D) values are in percentages in relation to the total emission or deposition of the month (stated in the last line).**

| mass [amu] | elemental composition | possible compound | April E | April D | May E | May D | June E | June D |
|---|---|---|---|---|---|---|---|---|
| 33.0335 | $C_1H_5O_1^+$ | methanol | | | 35.3 | 49.1* | 38.7 | 16.5* |
| 137.1325 | $C_{10}H_{17}^+$ | monoterpenes | 17.6 | 0 | 23.2 | 0 | 19.2 | 0 |
| 59.0491 | $C_3H_7O_1^+$ | acetone | 19.3 | 100* | 16.8 | 0 | 13.1 | <1* |
| 57.0699 | $C_4H_9^+$ | butene | 63.0 | 0 | 10.8 | 0 | 8.2 | 0 |
| 45.0335 | $C_2H_5O_1^+$ | acetaldehyde | | | | | 5.1 | 4.0* |
| 69.0699 | $C_5H_9^+$ | isoprene | | | 2.2 | <1* | 4.2 | 1.2* |
| 61.0284 | $C_2H_5O_2^+$ | acetic acid | | | | | 2.4 | 4.1* |
| 43.0178 | $C_2H_3O_1^+$ | fragment | | | | | 2.2 | 7.7* |
| 41.0386 | $C_3H_5^+$ | fragment | | | 2.6 | 3.7* | 2.3 | <1* |
| 31.0178 | $C_1H_3O_1^+$ | formaldehyde | | | | | <1 | 41.4 |
| 60.0471 | 60.0471 | unknown | | | 3.4 | 6.8* | 1.2 | 9.0* |
| 93.0699 | $C_7H_9^+$ | toluene | | | 2.4 | <1* | <1 | 1.8 |
| 69.0352 | 69.0352 | unknown | | | | | <1 | <1* |
| 67.0542 | $C_5H_7^+$ | cyclopentadiene | | | | | <1 | <1* |
| 70.0696 | 70.0696 | unknown | | | <1 | 2.6 | <1 | 1.7* |
| 99.0201 | 99.0201 | unknown | | | | | <1 | 3.6* |
| 84.9500 | 84.9500 | unknown | | | | | <1* | 4.8 |
| 95.0491 | $C_6H_7O_1^+$ | phenol | | | | | <1 | 1.1* |
| 135.1168 | $C_{10}H_{15}^+$ | p-cymene | | | | | <1* | <1* |
| 46.0287 | $C_1H_4O_1N_1^+$ | formamide | | | 1.9 | 23.9 | | |
| 118.9456 | 118.9456 | unknown | | | <1* | 9.0 | | |
| 71.0491 | $C_4H_7O_1^+$ | MVK/MACR | | | <1 | 7.9 | | |
| total emission and deposition [nmol m$^{-2}$ s$^{-1}$] | | | 0.68 | -0.01 | 1.39 | -0.17 | 3.12 | -0.12 |

(*) Values under the limit of detection ($2\,\sigma_{ind}$). $\sigma_{ind}$ was calculated using the propagation of error formula and the standard deviation at the borders of the individual 30 min CCFs.



**Table 2: Comparison between different studies using a PTR-TOF with the EC method. The listed compounds are limited to the ones measured in Hyytiälä. Numbers in parentheses describe deposition and emission, respectively. All values are 24h averages, except in Kaser et al., (2013a), where 8h average daytime values are presented.**

| mass [amu] | elem. comp. | net flux [nmol m$^{-2}$ s$^{-1}$] | | | | |
| --- | --- | --- | --- | --- | --- | --- |
| | | this study (21 days June) | Schallhart et al. (2016) | Park et al. (2013)[a] | Brilli et al. (2015) | Kaser et al. (2013a)[b] |
| 33.0335 | C$_1$H$_5$O$_1^+$ | 0.965* (-0.044/1.010) | 1.168 (-0.365/1.533) | 1.655 (-0.102/1.757) | 0.884 | 3.53[b] |
| 41.0386 | C$_3$H$_5^+$ | 0.050* (-0.001/0.051) | | 0.085 (-0.005/0.089) | | |
| 42.0338 | C$_2$H$_4$N$_1^+$ | 0.003 (-0.008/0.011) | 0.046 (-0.005/0.051) | | | |
| 43.0178 | C$_2$H$_3$O$_1^+$ | 0.027* (-0.011/0.038) | | 0.075 (-0.001/0.076) | | |
| 45.0335 | C$_2$H$_5$O$_1^+$ | 0.099* (-0.004/0.103) | 0.228 (-0.001/0.229) | 0.133 (-0.016/0.148) | 0.004 | 1.05[b] |
| 57.0699 | C$_4$H$_9^+$ | 0.199 (0/0.199) | | 0.016 (-0.011/0.027) | | |
| 59.0491 | C$_3$H$_7$O$_1^+$ | 0.297* (-0.001/0.297) | 0.335 (-0.01/0.345) | 0.281 (-0.004/0.286) | 0.035 | 0.13[b] |
| 60.0471 | unknown | 0.022* (-0.005/0.026) | | | | |
| 61.0284 | C$_2$H$_5$O$_2^+$ | 0.044* (-0.003/0.048) | 0.214 (-0.096/0.311) | 0.413 (-0.005/0.418) | | 1.64[b] |
| 67.0542 | C$_5$H$_7^+$ | 0.006* (-0.001/0.007) | | 0.012 (-0.004/0.017) | | |
| 69.0352 | unknown | 0.013* (-0.001/0.013) | | | | |
| 69.0699 | C$_5$H$_9^+$ | 0.083* (-0.003/0.086) | 6.466 (0/6.466) | 0.025 (-0.001/0.025) | 1.009 | 5.87[b] |
| 70.0696 | unknown | 0.007* (-0.001/0.008) | | | | |
| 89.0386 | C$_7$H$_5^+$ | 0.001 (-0.004/0.005) | | | | |
| 93.0699 | C$_7$H$_9^+$ | 0.020* (-0.001/0.021) | | 0.058 (0/0.058) | | |
| 99.0769 | unknown | 0.001 (-0.004/0.004) | | | | |
| 137.1325 | C$_{10}$H$_{17}^+$ | 0.430 (0/0.430) | 0.219 (-0.001/0.219) | 0.290 (0/0.290)[2] | 0.005 | 0.71[b] |
| Net flux in study | nmol m$^{-2}$s$^{-1}$ | 2.27 | 9.78 | 4.43 | 1.99 | 15.07[b] |
| Length of data set | days | 21 | 21 | 33 | 129 | 31 |
| Highest emitted compound | | methanol | isoprene | methanol | isoprene | MBO & isoprene |
| # of compounds with flux | | 17 | 29 | 494 | 13 | 15[c] |

(*) Values for downward flux are under the respective limit of detection (2 $\sigma_{ind}$). $\sigma_{ind}$ was calculated using the propagation of error formula and the standard deviation at the borders of the individual 30 min CCFs. ([a]) Park et al. (2013) published the 24h values of the identified masses, therefore no comparison of the unidentified compounds was possible. ([b]) Kaser et al. (2013a) published average daytime fluxes (10:00 -18:00) and therefore their results cannot be directly compared with the 24h average values. ([c]) 8 of the 15 compounds were only recorded after a hail storm event.





**Table 3: Statistics of the major compounds of SLP and the EC flux measurements. The compound names are educated guesses (see Sect. 2.2 for possible interferences). The EC fluxes are in bold whereas the SLP fluxes are written in normal text. The fitting parameters describe the slope (upper value) and the intercept (lower value) of the linear model between the EC and the SLP fluxes. $N$ is the number of the data points for each compound. The numbers in parenthesis are lower and upper quartiles. The unit for the mean, median, intercept and quantile values is nmol m$^{-2}$ s$^{-1}$.**

| nominal mass | $R^2$ | mean | median | 5% and 95% quant. | fitting | $N$ |
|---|---|---|---|---|---|---|
| 33 (methanol)* | 0.374 | 0.447 | 0.204 (-0.177, 1.161) | -1.602, 2.920 | 0.810 ± 0.220 | 92 |
| | | **0.553** | **0.730 (-0.931, 1.910)** | **-2.107, 3.296** | 0.171 ± 0.306 | |
| 42 (acetonitrile) | 0.003 | -0.009 | -0.003 (-0.026, 0.005) | -0.052, 0.026 | 0.296 ± 1.545 | 55 |
| | | **-0.017** | **-0.050 (-0.103, 0.072)** | **-0.209, 0.228** | -0.015 ± 0.040 | |
| 45 (acetaldehyde) | 0.033 | 0.019 | 0.024 (-0.106, 0.131) | -0.382, 0.481 | -0.195 ± 0.310 | 49 |
| | | **0.063** | **0.114 (-0.116, 0.245)** | **-0.406, 0.507** | 0.067 ± 0.082 | |
| 59 (acetone) | 0.051 | 0.176 | 0.090 (0.004, 0.262) | -0.156, 0.696 | 0.242 ± 0.191 | 119 |
| | | **0.173** | **0.210 (-0.046, 0.373)** | **-0.469, 0.677** | 0.131 ± 0.074 | |
| 61 (acetic acid)*,a | 0.064 | 0.336 | 0.244 (0.059, 0.504) | 0.003, 1.052 | 0.116 ± 0.130 | 49 |
| | | **0.025** | **0.058 (-0.029, 0.093)** | **-0.115, 0.195** | -0.014 ± 0.061 | |
| 69 (isoprene) | 0.127 | 0.082 | 0.033 (0.005, 0.091) | -0.009, 0.383 | 0.274 ± 0.160 | 82 |
| | | **0.035** | **0.035 (-0.013, 0.052)** | **-0.054, 0.104** | 0.013 ± 0.025 | |
| 93 (toluene) | 0.089 | 0.138 | 0.081 (0.024, 0.209) | -0.063, 0.519 | 0.088 ± 0.061 | 88 |
| | | **0.027** | **0.037 (-0.013, 0.052)** | **-0.054, 0.104** | 0.015 ± 0.014 | |
| 137 (monoterpenes) | 0.364 | 0.282 | 0.208 (0.084, 0.365) | 0.023, 0.754 | 0.503 ± 0.123 | 116 |
| | | **0.261** | **0.225 (0.135, 0.349)** | **0.019, 0.583** | 0.118 ± 0.051 | |

(*) sensitivity was derived from the instrumental transmission curve for the PTR-Quad. (a) the acetic acid sensitivity was estimated





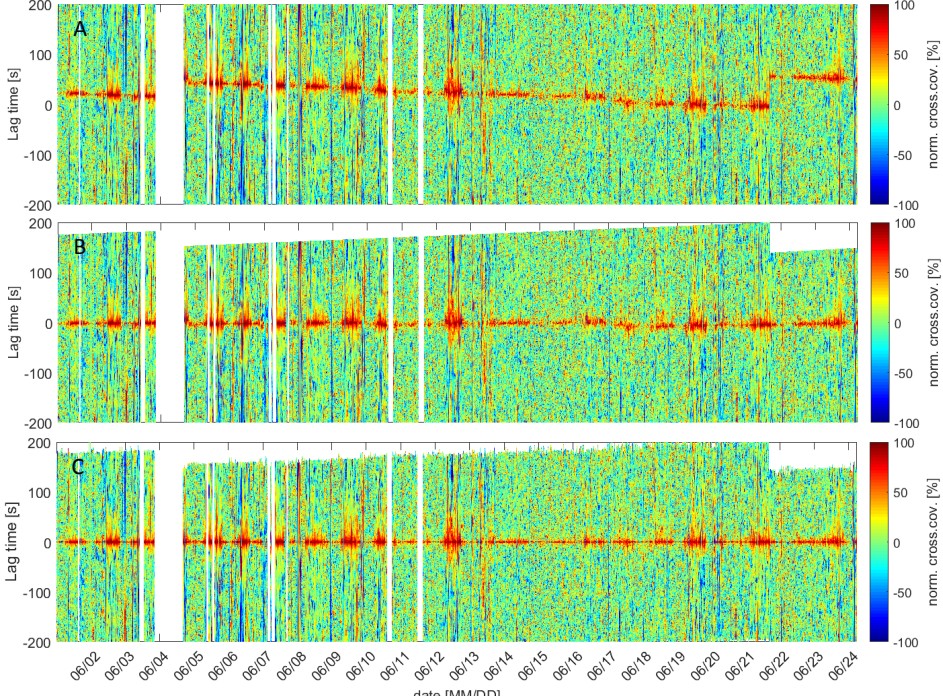

**Figure 1: A) Normalized CCFs for monoterpene measurements in June, without time shift correction. The shift between the two computer clocks is clearly visible. B) CCFs after correcting for lag time shifts. C) The CCFs of monoterpenes after the final lag time correction. In the final (third) step, the smoothed maximum of the CCF function (see Fig. 2) was sought for each compound and 30 min data individually, in a ± 10 s window of the previous lag time (step 1 & 2).**




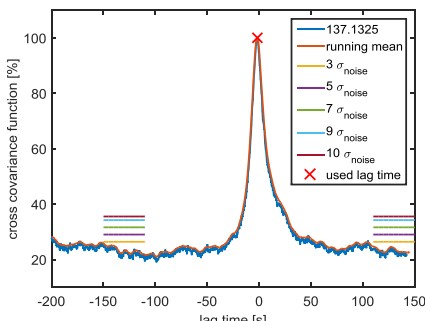

**Figure 2: For each compound which had a maximum above 3 $\sigma_{noise}$, the deviation of the CCF maxima from zero was used as the lag time correction.**





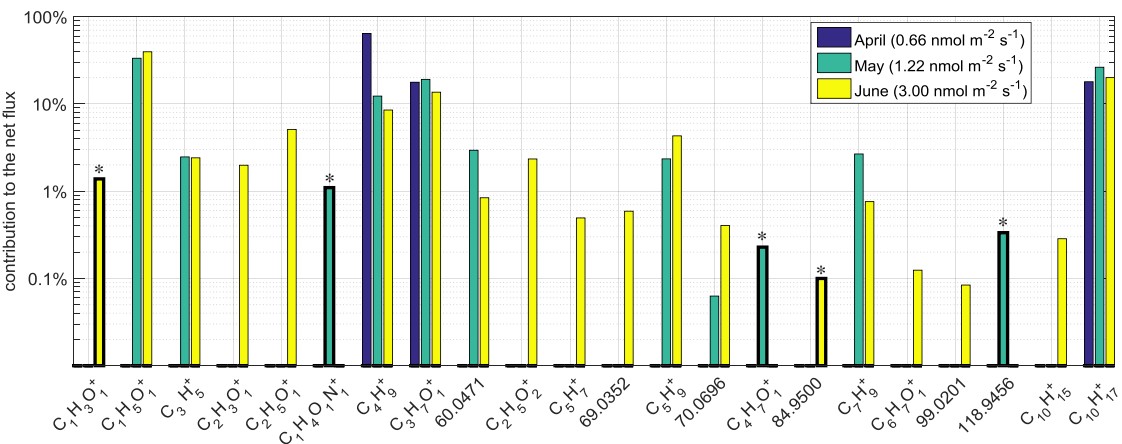

**Figure 3: Contribution of each compound to the 24h average net flux of the respective month. Bars with thick outlines and \* above them correspond to negative fluxes, where the absolute value was taken before plotting them in the logarithmic scale. See Table 1 for corresponding compound names.**





**Figure 4: Diurnal pattern of the 9 compounds with the highest fluxes, the remaining compounds being summed up as 'other'. The panels show the fluxes for April (top), May (mid) and June (bottom). The number of data points per hour is dependent on the quality criteria filtering and whether it is in an hour when the automatic background was measured.**




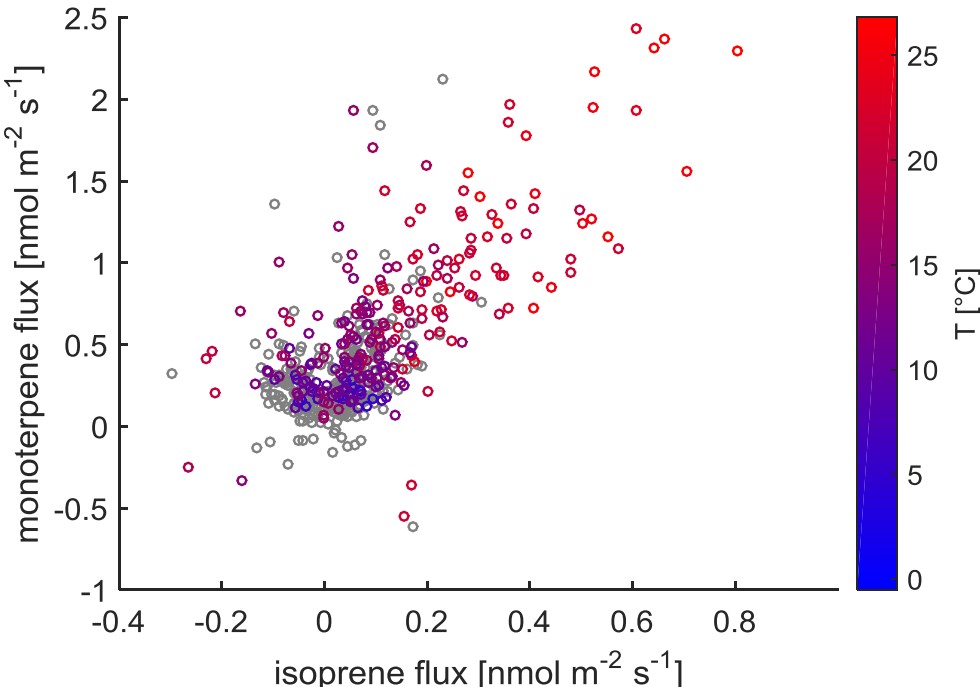

**Figure 5: Scatterplot of 30 min isoprene and monoterpene flux. The gray data points are values where the PAR was smaller than 200 µmol m$^{-2}$ s$^{-1}$. Isoprene fluxes are very low and especially during low PAR conditions they are heavily affected by noise and a mirroring effect (Langford et al., 2015).**





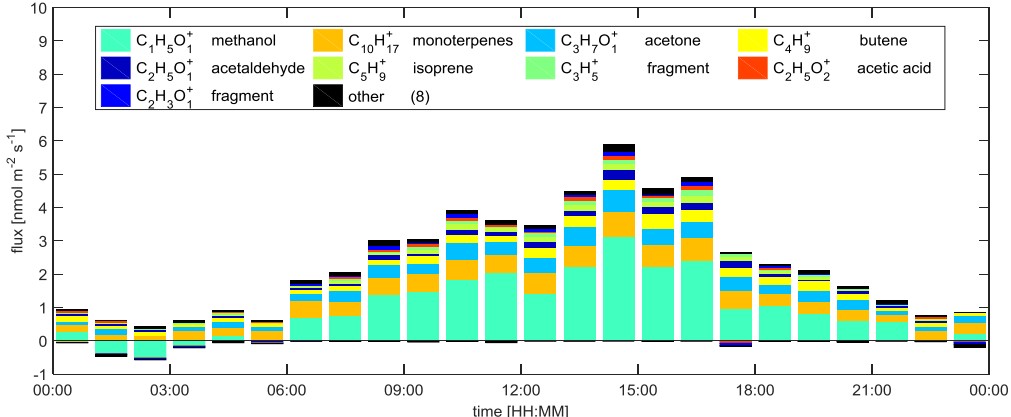

**Figure 6: Diurnal variation of the nine most emitted compounds during the 21 days of measurement in June. The remaining compounds are summed up as 'other'. The high variation in the flux seen in Fig. 4 (diurnal plot of 9 days of measurements in June) is reduced, as meteorological events (e.g. rain) have less impact on the result.**





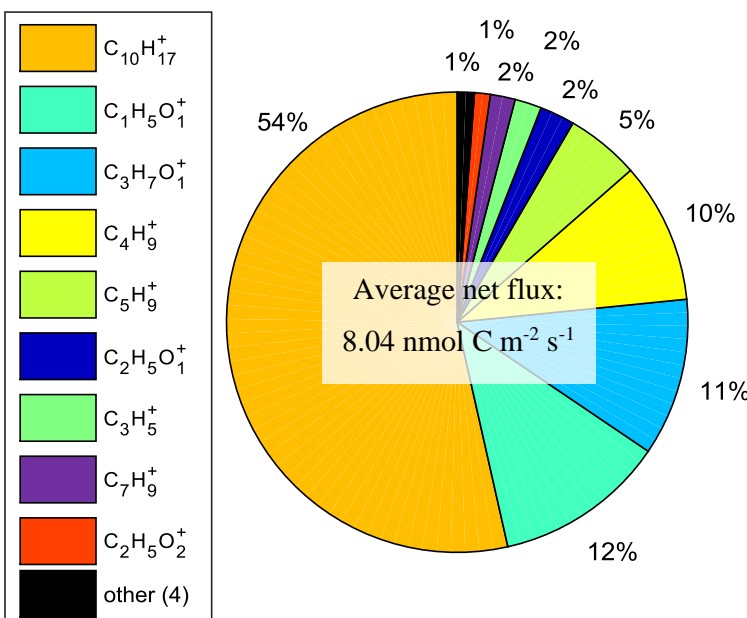

**Figure 7: Average net flux of the major carbon emitters. Compounds whose elemental composition could not be identified were disregarded. See Table 1 for the corresponding compound names.**





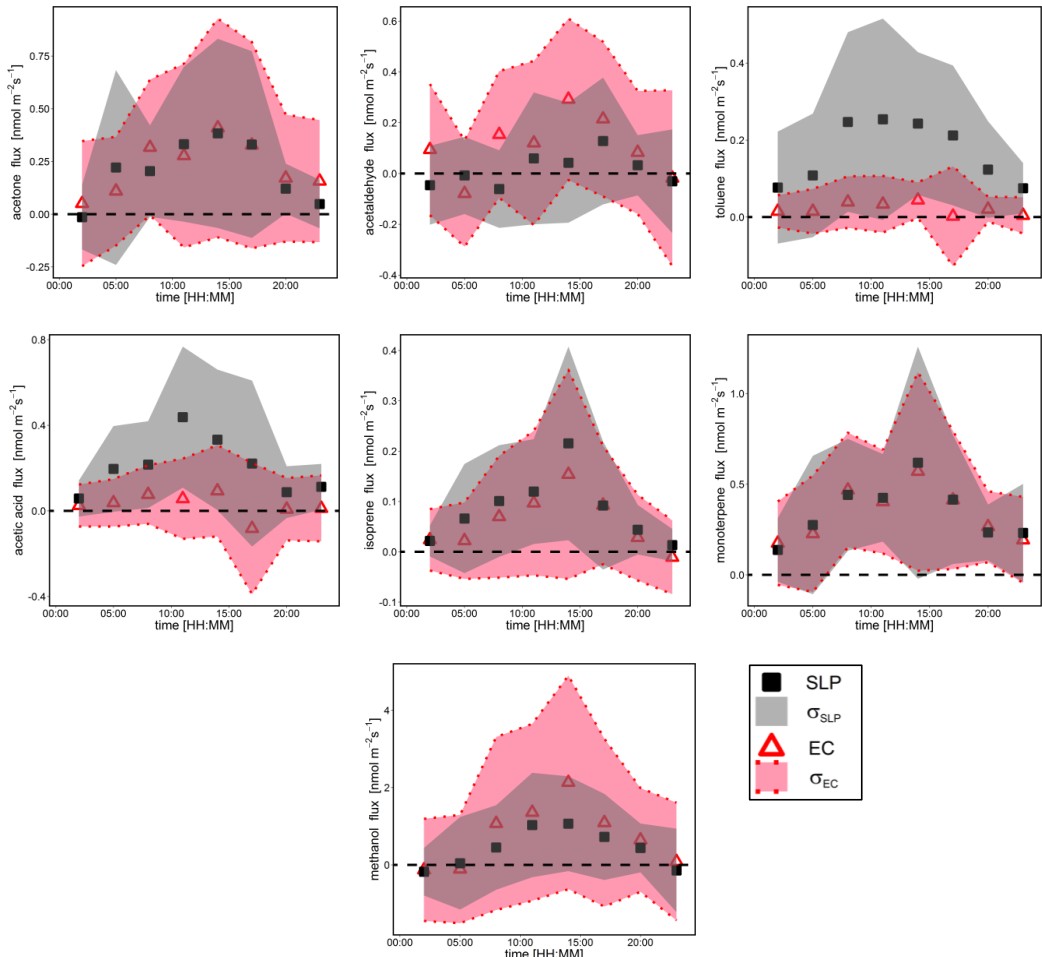

**Figure 8: Diurnal mean and standard deviation of the eddy covariance (red triangles) and the surface layer profile (black squares) flux of the major compounds measured by EC and SLP. The data is from April to the end of June 2013.**





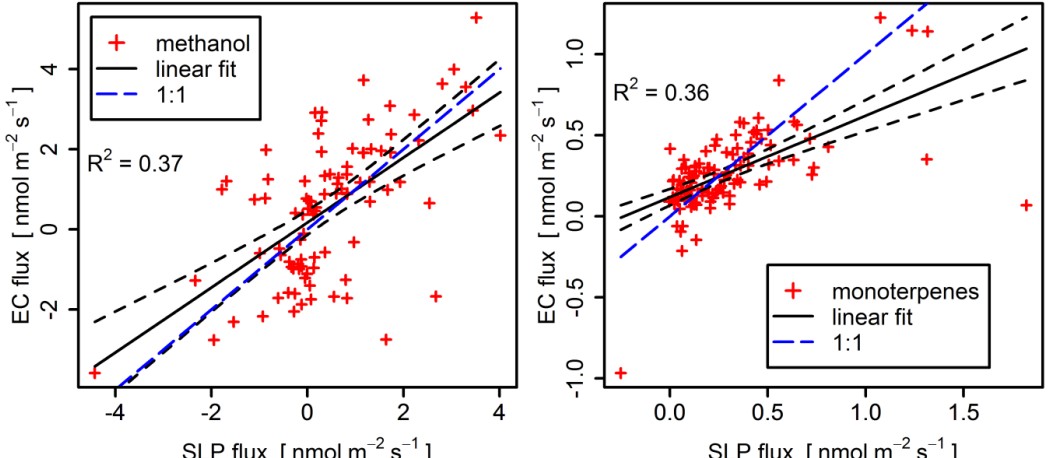

**Figure 9: The eddy covariance fluxes against the surface layer gradient fluxes (methanol and monoterpenes, April-May 2013). In addition to the actual scatter plots, the figures include linear fits (black solid lines) with confidence intervals (black dashed lines) and $R^2$ parameters.**





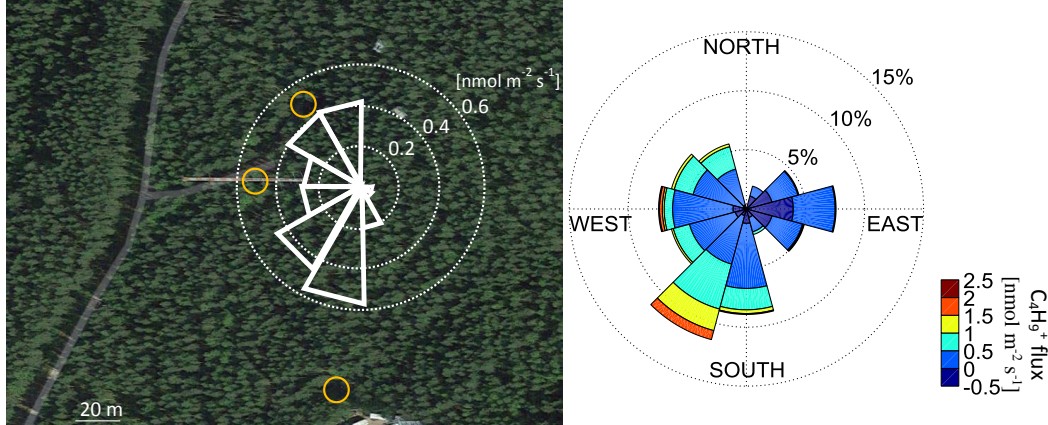

**Figure 10: On the left side the average C$_4$H$_9^+$ flux is shown (data of the whole campaign) in white. The orange circles illustrate the locations of the butanol using aerosol instruments. The average fluxes from north to south-southeast are under 0.05 nmol m$^{-2}$ s$^{-1}$. The map was taken from Google Maps (Imagenary©2016 Google, Map data©2016 Google). The windrose of the C$_4$H$_9^+$ fluxes is shown on the right.**



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
