# Peer review of "Temporal variation of VOC fluxes measured with PTR-TOF above a boreal forest"

_Atmospheric Chemistry and Physics, 2017_

## Referee Comment (RC1) · Anonymous Referee #1 · 21 Jun 2017

The paper by Schallhart et al. describes NMVOC flux measurements above a forested site in Finland using two PTR instruments. They calculate fluxes based on the direct eddy covariance and the indirect gradient method.

1) It is interesting that a much smaller number of actively exchanged VOCs were observed during this study than by Park et al., 2013, using similar methods for analyzing the entire mass spectrum. Is there an explanation? Was it a problem of the analysis (e.g. shifting time synchronization) or are Boreal forest generally just low (or not very active) BVOC emitting ecosystems? It appears that the measured net-fluxes are generally quite low when comparing with similar measurements from temperate and tropical forests.

2) No flux footprint analysis is presented. Why? It would seem important for the interpretation of the results, especially for a site which is largely comprised of a managed forest (with clearings). I also see there is a nearby lake. Knowing the footprint might also be a very relevant issue for the comparison between the two flux methods. The analysis would also seem to be crucial for the interpretation of butanol contamination from aerosol measurements (section 3.4). The plotted wind rose suggests that a large proportion of the wind-direction is from the S-W sector, where the buildings of the research station are situated.

3) m/z 93: While this mass is commonly shown to be associated with toluene using PTR ionization, there is evidence that a fragment from cymene can exhibit some interference (e.g. 20-30%) above evergreen forests. Have the authors verified that this is not the case – e.g. by correlating fluxes or concentrations of the corresponding ions? I also wonder whether benzene fluxes were detectable, and whether benzene to toluene (concentrations and/or fluxes) ratios could indicate local pollution.

4) Section 3.2. While being a valuable section, a number of studies are missing here – it would benefit the manuscript to conduct a more thorough literature review.

5) Table 2: 24 h flux average are chosen quite randomly and are often problematic, because most of the studies do not apply corrections to night time fluxes (e.g. storage and advection). A couple of clarifications would be important. Which (if any) corrections were applied to night time fluxes? Also why compare a 24h average with an 8h average – this does not make much sense. One should average the current dataset over the same period as each of the referenced datasets to be a meaningful comparison.

6) Page 10 Line 32: So if you conclude that the SMEAR station generally suffers from widespread butanol contamination from aerosol instrumentation, why talk about a 'butene flux' in this paper?

There are a large number of grammatical errors or confusing statements– I suggest copy-editing if this cannot be easily resolved by the authors. A small number of examples is listed below.

Page 5 Line 28: Overall 22 compounds with flux – change to "with a flux"

Page 5 Line 29: "C1" is an odd way to present a sum formula – I suggest to change it to "C" throughout the manuscript

Page 6 Line 4: measured in mid-afternoon – change to "during mid-afternoon"

Page 6 Line 6: methane is typically not considered a VOC – I suggest changing VOC to NMVOC throughout the manuscript as this is a more commonly used expression

Page 6 Line 6: The entire sentence here is confusing – I suggest to rephrase it.

Page 6 line 9: "for all months"

Page 6 line 10: "detectable fluxes"

Page 6 line 20: The heaviest measured compounds with detectable flux was - > "were"

Page 9 line 4: using surface layer profile method - > using "THE" surface layer method

Page 9 line 7: "ON" the same order

Page 9 line 8 : between the two measurements ?

Page 9 Line 12: "by" both methods

Page 11: "air chemistry"

Table 1 and 2, Figure 3, 6, 7: change "C1, O1, N1, etc." in the elemental composition column to "C, O, N"

---

## Referee Comment (RC2) · Anonymous Referee #2 · 14 Aug 2017

The paper written by Schallhart et al. describes VOCs fluxes above a boreal forest by using PTR-TOF and PTR-Quad instruments coupled with eddy covariance technique and indirect gradient method, respectively.

1) The fluxes are measured during 9 days in April and May and 21 days during June. I doubt, if only 9 days of months are enough to represent the whole month.

2) Page 2 line 34: Surroundings are described in paper published in 2005 and 2010. Please add at least current (2013) leaf area index, tree density, when the trees were planted or how old are the trees.

3) Page 2 line 35: I would not use "climatological mean temperature" but rather mean annual air temperature. From which years is the mean computed should be added.

Similarly for precipitation. Additionally the mean should cover the measured period. As there is (page 2 line 36) Pirinen et al., (2012 cited), it seems that the mean is only until 2012. However you measured in 2013.

4) Page 3 Line 9: There should be: data "were" analysed

5) Page 3 Line 28: Since here the rest should be in discussion section. It has no relationship to the measurement setup.

6) Page 4 Line 40: When your lines were heated and the effect was negligible, why to write about that? I would omit those sentences on lines 37-40.

7) Page 5 Lines 5-6: It is clear that the response time cannot be used in other studies. Do not need to be mentioned.

8) Page 5 Line 15 and afterwards: How could butane emissions dominated the flux, when the emissions are possibly originating artificially nearby the tower? This is a big problem and you should omit all the measured butane flux in here and in all the tables and graphs, since it is not product of the forest, as you conclude at page 10 line 31. Similarly, how you could calculate the C release caused by BVOCs, when you count artificial butane in there? Please recalculate all the values, where butane was used and all the graphs (figure 3, 4, 6, 7) and tables (table 1). The discussion section and results should be rephrased based on the new data without the butane flux.

9) Page 7 Line 30: Check supplement of M. Müller et al. (2015), doi:10.5194/acpd-15-31501-2015-supplement. The 99.0201 amu could be $C_4H_3O_3$.

10) Page 7 Line 32: 89.0386 amu could be $C_3H_5O_3$.

11) Page 8 Line 8: I do not see sense in comparing 24 h flux, measured in different periods of year at different plant ecosystems. Authors conclude that 8 hours flux by Kaser et al. (2013a) could not be compared to the 24 h net fluxes. Then why to put 8 h flux in Table 2?

12) Page 8 Line 21: Toluene fragment contribute to the carbon flux. That would mean, that toluene is produced by vegetation, however that is no longer discussed in all manuscript. Authors should bring more convincing results about toluene being released by Scots pine forest or give explanation how can be toluene emitted being anthropogenic. Check also figure 4, where is toluene emission and recalculate that.

13) Page 8 Line 22-25: I would add the study conducted by Juran et al. (2017), doi: 10.1016/j.agrformet.2016.10.005, since they identified portion of carbon released by BVOC to GPP on Norway spruce forest. I find that more close to forest in Finland than the mentioned papers here dealing with other ecosystems covered by very different trees.

14) Page 9 Line 9: Why there is such vertical difference between the two inlets? The footprints are very different than and thus the results are not comparable.

15) Page 9 Line 18: How is possible, that PTR-TOF has not detected those three compounds and less sensitive PTR-Quad has?

16) Page 9 Line 22: The correlation between monoterpene and methanol fluxes with each other at Fig 9 is rather bad than good. Similarly, R2 in Table 3 for acetonitrile, acetaldehyde, acetone and toluene suggests that there is no correlation at all. I doubt that the data were post-processed correctly or the instrument was working fine. Could you give explanation about that?

17) Page 9 Line 9: It might be useful to show transmission curve to check how certain is to calculate the sensitivities.

18) Table 3: Rephrase "educated guesses"

---

## Referee Comment (RC3) · Anonymous Referee #3 · 15 Aug 2017

Schallhart and colleagues present VOC flux measurements at the SMEAR II station in Southern Finland. The measurements reported are interesting to the biogenic VOC flux community, however, some more work needs to be done to be able to accept this manuscript for publication. In addition, the language should be edited throughout the manuscript in order to improve sentence structure and thus text readability. Some issues that deserve attention are shown in the list below.

P1L13 Correct 'only a three' to 'only three'

P1L1-2 This sentence needs some English editing

P2L5 "seems" should be "seem"

P3L37-39 rephrase

[Figure]

P4L17-25 It is not clear to me what is the advantage of the second step, because the second and third steps seem almost the same thing. Please clarify the reason why those steps were taken.

P5L6 Can the authors list the values of average correction factors for day and night separately?

P5L19 why the period of 21 days from 04 to 24 May is shown when only a period of 9 days was used?

P6L25-26 confusing sentence

P6L24-P7L34 Sections 3.1.2 and 3.1.3 should be rewritten to emphasize readability

P7L10 does this "highest total emission" refer to the maximum "average daytime emission"? Please clarify since it is difficult to discern from the (higher) 9.69 nmol/m2s shown on P7L13.

P7L36-P8L38 There are many other recent studies that, either with PTR-TOF or PTR-Quad, reported VOC fluxes that would give a more complete view to your discussion. Some of these studies report what fraction of the assimilated carbon was released as VOCs. In addition, some studies report VOC fluxes from monoterpene-emitting conifer species, which may be more relevant to your study. Some examples:

- McKinney et al (2011) https://doi.org/10.5194/acp-11-4807-2011

- Misztal et al (2011) https://doi.org/10.5194/acp-11-8995-2011

- Kalogridis et al (2014) https://doi.org/10.5194/acp-14-10085-2014

- Seco et al (2015) https://doi.org/10.1111/gcb.12980

- Juráň et al (2017) https://doi.org/10.1016/j.agrformet.2016.10.005

- Seco et al (2017) https://doi.org/10.1016/j.agrformet.2017.02.007

P9L5 Should the text "12 more compounds" be interpreted as 12 compounds exhibiting

SLP fluxes not found with the TOF-only EC analysis?

P9L22 Figure 8 shows indeed that fluxes for monoterpenes, isoprene and acetone were comparable. However, Table 3 lists the slope of the fit between the fluxes calculated with both methods and the slopes for all compounds (including monoterpenes) except acetone are far away from 1. The authors should further discuss this discrepancy, in addition to what is already discussed in P10L4-10. Also please rephrase the last sentence because from the text it seems that monoterpene and methanol were correlated with each other, when Fig 9 shows that what was correlated were the fluxes calculated with EC and SLP.

P10 section 3.4 the contamination from butanol used in CPCs is a major problem for the attempt to report butene fluxes. Since the authors have concluded that most of the signal of CH4H9+ is from butanol, the results and discussion section relating to butene should be omitted. I would recommend still having part of the manuscript dedicated to explaining what happened with butanol so other researchers can keep this contamination in mind for future experiments. Also, have the authors quantified the capacity of the forest (e.g. the fluxes measured on the eastern side of the forest) to take up butanol?

P10L28 How many compounds showed fluxes? Here and in section 3.2 the authors say 25, but in section 3.1 they say 22. Please be consistent.

P11L6-8 A good reference for this final sentence is Rinne et al 2016 https://doi.org/10.1016/j.atmosenv.2016.02.005

Fig 10 caption: "butanol using aerosol instruments" is confusing
* * *

---

## Author Comment (AC1) · 20 Oct 2017

We would like to thank the reviewers for their detailed and helpful comments and suggestions, which improved the manuscript. The comments of the reviewers are in black, while our replies are marked in red. The revised manuscript is attached after the comments and answers to all reviewers.

**Anonymous Referee #1**

Received and published: 21 June 2017

The paper by Schallhart et al. describes NMVOC flux measurements above a forested site in Finland using two PTR instruments. They calculate fluxes based on the direct eddy covariance and the indirect gradient method.

1) It is interesting that a much smaller number of actively exchanged VOCs were observed during this study than by Park et al., 2013, using similar methods for analyzing the entire mass spectrum. Is there an explanation? Was it a problem of the analysis (e.g. shifting time synchronization) or are Boreal forest generally just low (or not very active) BVOC emitting ecosystems? It appears that the measured net-fluxes are generally quite low when comparing with similar measurements from temperate and tropical forests.

A good question, we do not suspect the analysis being the cause of the discrepancy in the number of actively exchanged VOCs. The analysis was same as in Schallhart et al., 2016, where VOC fluxes with constant lag time (2.6 s) were measured above a very active (24 h net flux: 9.78 nmol  $m^{-2} s^{-1}$ ) oak forest. Still in the oak forest, only 29 compounds were actively exchanged. Schallhart et al. (2016) also gives a more detailed explanation of automated method and compares this method with a manual method.

Therefore, the reason in the huge difference between 494 (Park et al., 2013), 29 (Schallhart et al., 2016) and 17 (for 21 days in June; this study) exchanged compounds must be primarily caused by the behavior of the orange yard, which seems to be a unique ecosystem. The measurement setup, e.g. shorter inlet lines as used by Park et al. (2013), can reduce the losses for heavier, stickier and/or more reactive VOCs, but cannot explain an order of magnitude difference in number of exchanged compounds. Additionally, the meteorological circumstances (PAR, T) can lead to stress responses in plants and higher photochemistry could lead to additional oxidation products.

Overall, a comparison between different ecosystems is very challenging. The flux of individual compounds are similar as in previous boreal forest measurements indicating that the measurements of ecosystem fluxes worked well.

If the oak forest and the boreal forest are compared, the difference in number of actively exchanged compounds should be mainly caused by the lower emissions in the boreal forest and fewer compounds are released in detectable amounts. The shift in lag time should have a minor effect (after the described corrections).

2) No flux footprint analysis is presented. Why? It would seem important for the interpretation of the results, especially for a site which is largely comprised of a managed forest (with clearings). I also see there is a nearby lake. Knowing the footprint might also be a very relevant issue for the comparison between the two flux methods. The analysis would also seem to be crucial for the interpretation of butanol contamination from aerosol measurements (section 3.4). The plotted wind rose suggests that a large proportion of the wind-direction is from the S-W sector, where the buildings of the research station are situated.

Thank you for your comment, we agree that footprints would be important for interpreting the results. Therefore, we added average cumulative footprints (50% and 80%) for both the EC and the SLP measurements. The average footprints were calculated from the measurement period using daytime

values (07:00 - 19:00). During night, the flux footprint would be generally larger due to stronger stable conditions, but the VOC emissions are, on the other hand, usually low.

Cumulative footprints were estimated using an analytical function by Kormann and Meixner (2001). The algorithm uses horizontal wind speed, Obukhov length, standard deviation of a lateral wind, roughness length (estimated to be ca. 1.5 m), measurement height and zero displacement height (we used a value 13 m) to determine the footprint function. According to Horst (1999), footprints are similar for both the EC and the SLP methods if the measurement heights are equal. Therefore, we used the same algorithm also for the SLP fluxes.

Determined footprints have error sources and thus the results should be taken as estimates. The nominal measurement height of the SLP fluxes was 36 m (for unstable stratification). However, we used the height 33 m because horizontal wind measurements were available from that level. This underestimates slightly the footprints. On the other hand, Kormann & Meixner algorithm excludes an effect of roughness sublayer which leads to an underestimation of the turbulent transport and thus overestimates the cumulative footprints.

3) m/z 93: While this mass is commonly shown to be associated with toluene using PTR ionization, there is evidence that a fragment from cymene can exhibit some interference (e.g. 20-30%) above evergreen forests. Have the authors verified that this is not the case – e.g. by correlating fluxes or concentrations of the corresponding ions? I also wonder whether benzene fluxes were detectable, and whether benzene to toluene (concentrations and/or fluxes) ratios could indicate local pollution.

We agree that para-cymene can influence the signal and stated it in the end of Sect. 2.2. With the E/N used in this study, over 80% of the para-cymene signal should fragment to 93.0699 Da (Tani et al., 2003). Our intention was to name each signal depending on the mass where it was measured and discuss the possible compounds in Sect. 2.2. However, we realize that this can lead to misinterpretations and therefore will rename all signals measured at mass 93.0699 from toluene to toluene & p-cymene.

The correlation between the p-cymene (135.1168 Da) and toluene & p-cymene (93.0699 Da) concentration was 0.79. Due to the small fluxes of p-cymene (135.1168 Da), the 30 min files (used for the correlation) were influenced by noise, leading to a correlation with toluene & p-cymene (93.0699 Da) beneath 0.3.

Benzene fluxes were not detected and the average toluene & p-cymene/ benzene concentration ratio in June was 1.26 (standard deviation: 0.51). The toluene/benzene ratio from fresh anthropogenic plume is 4.25 (Warneke et al., 2007) and in the order of 2 for fresh traffic and the ratio goes down as the air mass is oxidized during transported from the source (Gelencsér et al., 2007) indicating that the source is not dominated by local anthropogenic or traffic sources.

4) Section 3.2. While being a valuable section, a number of studies are missing here – it would benefit the manuscript to conduct a more thorough literature review.

We thank the reviewer for this comment. The aim in this paper is to give an overview over the relative new and few already published studies, where no preselection of VOCs had to be made and, therefore, all VOCs with a flux above the detection limit (excluding instrument related signals) are reported. Thus the literature of VOC fluxes was focused on comparing EC fluxes measured with PTR-TOF that can scan the full range of VOCs emitted, instead of studies with a preselected set of studied compounds, typically 10-20 masses with the DEC fluxes using the PTR-Quad. Furthermore, we omitted publications which use less than four days of data (e.g. Ruuskanen et al., 2011) as fluxes would be prone to statistical variation and, therefore, be not representative of the ecosystem. Also studies which concentrate on specific VOCs are disregarded (e.g. Fares et al., 2013; Su et al., 2016).

Thus, as stated, we would like just to include EC fluxes (measured with PTR-TOF), as a comparison with DEC fluxes (measured by PTR-Quad) would be out of the scope of this study and would need, due to the large amount of studies, an individual review paper.

We added Juráň et al. (2017), where approximately five days of VOC fluxes were measured using EC with a PTR-TOF. However, we added the fraction of the assimilated carbon released from VOCs from Misztal et al. (2011) and Seco et al. (2017).

5) Table 2: 24 h flux average are chosen quite randomly and are often problematic, because most of the studies do not apply corrections to night time fluxes (e.g. storage and advection). A couple of clarifications would be important. Which (if any) corrections were applied to night time fluxes? Also why compare a 24h average with an 8h average – this does not make much sense. One should average the current dataset over the same period as each of the referenced datasets to be a meaningful comparison.

We agree that 24 h flux values are problematic, as the night time fluxes are more prone to errors. As suggested we added 8 h average flux values from this study, to compare to the values of Kaser et al. (2013) and clearly marked them as 8 h averages.

However, especially when measuring in northern countries, where emission are low but continue longer than in mid latitude areas, 8 h fluxes can lead to an underestimation. In high latitudes the sun sets only for a few hours in the night in summertime and temperature as well as light and temperature dependent emissions continue to midnight (maximum length of daytime light in this study: 19.5 h) and using midday 8 h flux averages would further decrease the cumulative emissions . We clarified in Sect. 2.3 that no additional corrections were applied to the measured fluxes (in addition to the high frequency attenuation correction).

6) Page 10 Line 32: So if you conclude that the SMEAR station generally suffers from widespread butanol contamination from aerosol instrumentation, why talk about a 'butene flux' in this paper?

Similarly as discussed in question 3, the initial idea was to always use the compound name of the measured mass and discuss the identification of the compound in Sect. 2.2. We agree that this can lead to misinterpretations and changed all butene references to butene & butanol. We would like to keep butene in the name, as a separation of the  $C_4H_9^+$  signal into butene and butanol fragment is, with our measurements setup, not possible. However, we clarify at several points that the butene & butanol emissions are anthropogenic and refer to Sect. 3.4.

There are a large number of grammatical errors or confusing statements– I suggest copy-editing if this cannot be easily resolved by the authors. A small number of examples is listed below.

We thank the reviewer for this comment the manuscript was checked carefully for mistakes after all reviewer comments were implemented.

Page 5 Line 28: Overall 22 compounds with flux – change to "with a flux"

Page 5 Line 29: "C1" is an odd way to present a sum formula – I suggest to change it to "C" throughout the manuscript

Page 6 Line 4: measured in mid-afternoon – change to "during mid-afternoon"

Page 6 Line 6: methane is typically not considered a VOC – I suggest changing VOC to NMVOC throughout the manuscript as this is a more commonly used expression

We agree with the reviewer, however, we would like to stay with the term VOC. Therefore the sentence was rewritten and any reference to methane was omitted.

Page 6 Line 6: The entire sentence here is confusing – I suggest to rephrase it.
Page 6 line 9: "for all months"
Page 6 line 10: "detectable fluxes"
Page 6 line 20: The heaviest measured compounds with detectable flux was -> "were"
Page 9 line 4: using surface layer profile method - > using "THE" surface layer method
Page 9 line 7: "ON" the same order
Page 9 line 8: between the two measurements ?
Was changed to: The discrepancy between the results of the two measurements is mainly [...]
Page 9 Line 12: "by" both methods
Page 11: "air chemistry"
Table 1 and 2, Figure 3, 6, 7: change "C1, O1, N1, etc." in the elemental composition column to "C, O, N"
All above mentioned examples were corrected.

Note: The unit of mass was changed from amu to Da, to ensure that the reader knows that the mass units used are based on the  ${}^{12}C$  atom.

method) could be generated from a random process, or if there was a real positive or negative flux. This was done for night (02:00 - 08:00) and daytime (11:00 - 17:00) values.

Therefore, the data for the classification if compounds were exchanged or not, differed between the two datasets. The SLP used a longer period, including June data from previous years with higher temperature, but the SLP fluxes were measured every third hour only. The EC fluxes were measured just in 2013, but continuously. This difference was necessary, as the SLP fluxes would otherwise have too little data for a classification (

Figure Z1: Mass dependency of the transmission of the PTR-Quad (left) and scatter of the measured and calculated normalized sensitivity.

**Anonymous Referee #3**

Received and published: 15 August 2017

Schallhart and colleagues present VOC flux measurements at the SMEAR II station in Southern Finland. The measurements reported are interesting to the biogenic VOC flux community, however, some more work needs to be done to be able to accept this manuscript for publication. In addition, the language should be edited throughout the manuscript in order to improve sentence structure and thus text readability. Some issues that deserve attention are shown in the list below.

P1L13 Correct 'only a three' to 'only three'

changed

P1L1-2 This sentence needs some English editing

We rephrased the sentence to: "The VOC exchange followed a seasonal trend and the emissions increased from spring to summer"

P2L5 "seems" should be "seem"

changed

P3L37-39 rephrase

We rephrased the section.

P4L17-25 It is not clear to me what is the advantage of the second step, because the second and third steps seem almost the same thing. Please clarify the reason why those steps were taken.

We thank the reviewer for this comment. After the first step, all lag times are similar than the lag time of the monoterpenes. The second step is used to correct compound specific lag times, e.g. if certain compounds would have on average a shorter or longer lag time, e.g. due to interaction with RH or tube walls (similar to the response time). This correction is constant over all the measurement period and ensures that the next step (three) uses a +- 10 s time window around the maximum. In step 3 every 30 min flux point individually is corrected. This corrects small fluctuations in the lag time, e.g. when the linear regression fit was not precise enough. We added a sentence explaining step 2 to the revised manuscript.

P5L6 Can the authors list the values of average correction factors for day and night separately?

Yes, we added the average daytime value of 16% (between 09:00 and 17:00) and nighttime value 23% (between 20:00 and 04:00).

P5L19 why the period of 21 days from 04 to 24 May is shown when only a period of 9 days was used?

During May we had technical problems with the 3d-sonic anemometer, which stopped recording data during several occasions during the period of 04 May 2013 to 24 May 2013. Therefore, only 423 30 min files (equals 211.5 h, approximately 9 days of data) of fluxes could be calculated in this period. We clarified this section in the manuscript.

P6L25-26 confusing sentence

We rewrote the sentence and improved the readability.

P6L24-P7L34 Sections 3.1.2 and 3.1.3 should be rewritten to emphasize readability

We rewrote the mentioned sections and improved the readability.

P7L10 does this "highest total emission" refer to the maximum "average daytime emission"? Please clarify since it is difficult to discern from the (higher) 9.69 nmol/m2s shown on P7L13.

We changed the "highest total emission" to "highest 24 h total emission". We did not use "average daytime emission", as the mentioned value includes also nighttime emissions.

P7L36-P8L38 There are many other recent studies that, either with PTR-TOF or PTRQuad, reported VOC fluxes that would give a more complete view to your discussion. Some of these studies report what fraction of the assimilated carbon was released as VOCs. In addition, some studies report VOC fluxes from monoterpene-emitting conifer species, which may be more relevant to your study. Some examples:

- McKinney et al (2011) https://doi.org/10.5194/acp-11-4807-2011
- Misztal et al (2011) https://doi.org/10.5194/acp-11-8995-2011
- Kalogridis et al (2014) https://doi.org/10.5194/acp-14-10085-2014
- Seco et al (2015) https://doi.org/10.1111/gcb.12980
- Juráň et al (2017) https://doi.org/10.1016/j.agrformet.2016.10.005
- Seco et al (2017) https://doi.org/10.1016/j.agrformet.2017.02.007

We thank the reviewer for the suggested literature and also refer to Reviewer 1 Q4:

As stated we would like to include just EC fluxes (measured with PTR-TOF), as a comparison with DEC fluxes (measured by PTR-Quad) would be out of the scope of this study and would need, due to the amount of studies, an individual review paper.

The aim in this paper was to give an overview over the relative new and few already published studies, where no preselection of VOCs had to be made and, therefore, all VOCs with a flux above the detection limit (excluding instrument related signals) are reported.

Furthermore, we omitted publications which use less than four days of data (e.g. Ruuskanen et al., 2011) as fluxes would be prone to statistical variation and, therefore, be not representative of the ecosystem. Also studies which concentrate on specific VOCs are disregarded (e.g. Fares et al., 2013; Su et al., 2016).

We added Juráň et al. (2017), where approximately five days of VOC fluxes were measured using EC with a PTR-TOF. However, as 24 h flux values were not stated for all measured compounds, the study was not added in Table 2. We also we added the fraction of the assimilated carbon released from VOCs from Misztal et al. (2011) and Seco et al. (2017).

McKinney et al. (2011) and Kalogridis et al. (2014) were not added, as no VOC/NEE carbon fraction was stated. Seco et al. (2015) was not used, as just the daytime fraction is stated.

P9L5 Should the text "12 more compounds" be interpreted as 12 compounds exhibiting SLP fluxes not found with the TOF-only EC analysis?

No, the opposite, there were 12 more compounds measured with the EC method that were not found with the SLP method. We clarified this in the text.

P9L22 Figure 8 shows indeed that fluxes for monoterpenes, isoprene and acetone were comparable. However, Table 3 lists the slope of the fit between the fluxes calculated with both methods and the slopes for all compounds (including monoterpenes) except acetone are far away from 1. The authors should further discuss this discrepancy, in addition to what is already discussed in P10L4-10. Also please rephrase the last sentence because from the text it seems that monoterpene and methanol were correlated with each other, when Fig 9 shows that what was correlated were the fluxes calculated with EC and SLP.

The mentioned major discrepancy is caused by the high intercept of the fitting. In the case of the monoterpenes, the intercept is around 50% of the mean flux, explaining the flat slope of 0.5. When we forced the offset to be zero, the slope increase to 0.7. The still existing discrepancy from unity can be partly explained by the outliers (Fig. 9), which are far off the 1:1 line and thereby decrease the slope (with their high influence on the explained sum of squares).

A similar case is the slope of isoprene. Here, the offset is about 30% of the mean flux and again, by forcing a zero intercept the slope increased. Furthermore, a close look at Fig. 8 shows, that there is clearly a smaller flux measured by the EC method, when compared with the SLP method and thereby the slope should be less than one.

Acetone has the highest intercept compared to the mean flux (> 60%) and also higher uncertainty of the slope  $(0.276 \pm 0.191)$  with respect to the compounds mentioned above.

When the offset was forced to be zero, the  $R^2$  values decreased for all compounds.

Overall, the discrepancy from unity can be explained by the scatter of the measured fluxes, which is shown in the uncertainties of the fitting parameters. The scatter is caused, as discussed in the paper, by variation in turbulence measurements, noise of the VOC measurements, different inlet heights (36 m, 23 m), methods (SLP, EC), instruments (PTR-Quad, PTR-TOF) and footprints (see revised manuscript Fig. 8). The comparison also lacked high flux periods, as can be seen when comparing the values of the monoterpene EC flux in Fig 8 (right side; 3 values above 1 nmol m-2 s-1) and Fig. 4 (bottom panel), where in the 9 day period already 44 values were above 1 nmol m-2 s-1 (maximum emission 2.44 nmol m-2 s-1).

We added a discussion about the discrepancy of the fitted slopes from unity to the manuscript.

P10 section 3.4 the contamination from butanol used in CPCs is a major problem for the attempt to report butene fluxes. Since the authors have concluded that most of the signal of CH4H9+ is from butanol, the results and discussion section relating to butene should be omitted. I would recommend still having part of the manuscript dedicated to explaining what happened with butanol so other researchers can keep this contamination in mind for future experiments. Also, have the authors quantified the capacity of the forest (e.g. the fluxes measured on the eastern side of the forest) to take up butanol?

See also the answer to Reviewer 2 Q8:

We agree with the reviewer, that the anthropogenic nature of the butene & butanol flux is lost, if just parts of the manuscript are read. Therefore we will omit the artificial butene & butanol flux in the total emission (Table 1 and 2), mentioned net fluxes and in Fig. 7. Still, the individual values will be presented in Table 1 and 2 as well as in Figs. 3, 4 and 6 as we would like to inform the reader that these fluxes do exist. All values connected with butene & butanol were labelled as anthropogenic to clarify that the flux does not come from the same biogenic source as other VOCs.

In the Fig. 10 left panel a small deposition of butene & butanol has been found with -0.04 nmol m-2 s-1 (0°-30° sector, NNE) and < 0.01 nmol m-2 s-1 (30°-60°, NE) sector. However, if this is active uptake of the forest or dry deposition we cannot answer with our measurement setup.

P10L28 How many compounds showed fluxes? Here and in section 3.2 the authors say 25, but in section 3.1 they say 22. Please be consistent.

We thank the reviewer for the comment and clarified the text. Overall, 25 compounds with fluxes have been discovered, 17 during the 21 day measurements in June, and 22 during the 3 periods in April, May and June. We rephrased the sentence in Sect. 3.1.

P11L6-8 A good reference for this final sentence is Rinne et al 2016 https://doi.org/10.1016/j.atmosenv.2016.02.005

We added the reference.

Fig 10 caption: "butanol using aerosol instruments" is confusing

Was changed to "[...] aerosol instruments, which use butanol."

Note: The unit of mass was changed from amu to Da, to ensure that the reader knows that the mass units used are based on the  ${}^{12}C$  atom.

**Temporal variation of VOC fluxes measured with PTR-TOF above a boreal forest**

Simon Schallhart1, Pekka Rantala1, Maija K. Kajos1, Juho Aalto2, Ivan Mammarella1, Taina M. Ruuskanen1 and Markku Kulmala1

1Department of Physics, University of Helsinki, Finland 2Department of Forest sciences, University of Helsinki, Finland

Correspondence to: Simon Schallhart (simon.schallhart@helsinki.fi)

Abstract. Between April and June 2013 fluxes of volatile organic compounds (VOCs) were measured in a Scots pine and Norway spruce forest using the eddy covariance (EC) method with a proton transfer reaction time of flight (PTR-TOF) mass spectrometer. The observations were performed above a boreal forest at the SMEAR II site in southern Finland. We found a total of 25 different compounds with exchange and investigated their seasonal variations from spring to summer. The majority of the net VOC flux was comprised of methanol, monoterpenes, acetone and butene & butanol. The butene and butanol emissions were concluded to not originate from the forest and, therefore, be anthropogenic.- The VOC exchange followed a seasonal trend and the emissions increased from spring to summerThe 
[revised manuscript text omitted]
 step corrects possible differences in the average lag time between the compound of interest and the monoterpenes. was This lag time was calculated for each month and each compound separately.
- III. The third and last step was used to correct for smaller shifts, in case the first correction, with the linear regression, was not precise enough. Therefore, each individual 30 min CCF was smoothed by a running mean ( $\pm 24$  step averaging) and the location of the maximum in a  $\pm 10$  s time-window around the previously calculated lag time was recorded (Taipale et al., 2010) as shown for the monoterpenes in Fig. 1c.

To classify how many of the hundreds of measured compounds show an exchange, a method described in Park et al., (2013) and Schallhart et al., (2016) was used. This method compares the maximum of the averaged, absolute CCF with a certain noise threshold. To reduce the impact of noise, the average, absolute CCF was smoothed ( $\pm 12$  step averaging) and the location of the maximum in a  $\pm 10$  s time window detected. This position was used in the average, absolute CCF (not smoothed) and compared with the  $\sigma_{noise}$  (standard deviation of the noise). The  $\sigma_{noise}$  was calculated for 60 s at the borders

of the average absolute CCF. If the ratio between the calculated maximum and the  $\sigma_{noise}$  was higher than three, the compound was classified to have detectable flux (Fig. 2). This method was applied to flux determinations for each month separately.

The flux underestimation caused by high frequency attenuation was estimated using a parametrization described by Horst (1997). The method uses a system response time and information about stability and horizontal wind speed for estimating the attenuation. The system response time was determined to be around 1.2 s for monoterpenes and the same response time was also used for all the other compounds. However, the response time of e.g. water soluble compounds might be larger due to desorption and absorption processes on the tube walls, as a function of relative humidity and sampling line aging (Mammarella et al, 2009; Nordbo et al., 2013 and 2014). This potentially causes errors of a few percentage on the flux values. However, we expect the effect to be reduced, because a heated sampling tube was used. The low frequency corrections that are based on theoretical transfer function shapes (e.g. Rannik and Vesala, 1999), were not applied in this study.

One should note that the determined response time describes the flux attenuation of the whole measurement setup, including the tubing, a horizontal separation between the inlet and the anemometer and the instrument itself. Thus, the response time of this study cannot be used in the case of other PTR TOF measurements at other locations. In this case, the average attenuation factor was 18%. On average, the correction factor was smaller during the day (16%; 09:00 to 17:00) and larger at night (23%; 20:00 to 04:00).

No additional corrections were applied to the measured fluxes (e.g. storage correction).

**2.4 Flux quality criteria**

The measured fluxes were filtered by three quality criteria, to reduce the systematic uncertainty and ensure their representativeness:

First, the data were flagged if the tilt angle, resulting from the coordinate rotation of sonic anemometer wind velocity components (Kaimal and Finnigan, 1994), was more than 5°, which was the case for 11.9% of the data. Second, all 30 min records with a friction velocity less than 0.2 m s-1 were flagged. Following this, 11.2% of the data was-were flagged. Finally, the flux steady-state test was applied according to Foken and Wichura (1996). All flagged flux values were removed from further analysis. The rejection rate between April and June was 34.1%, 35.1% and 30.5% for acetone, butene & butanol and the monoterpenes, respectively. For the monoterpenes, the rejection rates for the measurement periods in April, May and June were 19.1%, 17.6% and 30% (daytime) and 25.6%, 24.0% and 43.7% (nighttime).

**2.5 Flux selection**

The temporal behavior of the VOC exchange was investigated by measuring periods in April, May and June 2013. These periods give insight to the VOC exchange during snowmelt, start of the growing season and summer. VOC emissions from April, May and June 2013 were selected for a monthly comparison. Because of technical problems with the anemometer-, which stopped recording data on several occasions during the start of the growing season, only  $423 \times 30$  min files (~9 days of data) of VOC fluxes could be calculated in the period from 04 May 2013 to 24 May 2013 in May, only a nine day period, from 04 May 2013 to 24 May 2013, could be used. The standard deviation of noise in the averaged, absolute CCFs ( $\sigma_{noise}$ ) determines the exchange threshold and is directly dependent on the amount of data. Therefore, the same amount of data ( $423 \times 30$ \_min files) was selected to represent each month-period from snowmelt to summer and make thea comparison between those periods with the other months possible. For all three monthsperiods, the absolute

mean of the CCFs between 10:00 and 16:00 (UTC+2) was used to find compounds with statistically significant flux (Park et al., 2013; Schallhart et al., 2016). For Aprilthe snowmelt, the measurements were from 14 April 2013 to 24 April 2013 and for June-summer the hottest period was selected, from 01 June 2013 to 12 June 2013.

**3. Results and Discussion**

The PTR-TOF measures the mass of the VOCs in the ambient sample, which can be used to calculate the elemental composition. Therefore, no structural information of the measured molecules is known and the identification of compounds relies on literature and gas chromatograph measurements. Major masses affected by fragmentation and compounds with high uncertainties are discussed in the following.

The mass 69.0699 Da with the elemental composition  $C_5H_{9^+}$  was called isoprene & MBO, as 2-methyl-3-buten-2-ol (MBO) fragmented to this mass and had a substantial influence on the signal (e.g. Kaser et al., 2013b). Similarly, the mass 93.0699 Da with the elemental composition  $C_7H_{9^+}$  was called toluene & p-cymene, as p-cymene fragments were suspected to affect the signal (Tani et al., 2003). Formaldehyde has only a slightly higher proton affinity compared with the primary ion and therefore back reactions, which are humidity dependent, from protonated formaldehyde to water occur (de Gouw and Warneke, 2006; Inomata et al., 2008; Vlasenko et al., 2010). This may have led to an artificial flux, which was caused by water vapor fluctuations. Therefore, the formaldehyde fluxes are very uncertain. The signal at mass 57.0699 with a protonated composition of  $C_4H_{9^+}$  was called butene & butanol, as butanol was suspected to be caused by the aerosol instrumentation, they were disregarded in all reported net fluxes and total emissions. The monoterpenes ( $C_{10}H_{17}^+$ ) were measured at mass 137.1325 Da only. The monoterpene fragment at mass 81.0699 Da ( $C_6H_9^+$ ) was identified by its Pearson correlation of 0.99 (30 min integrated data) with the signal at 137.1325 Da and was disregarded from further analysis.

**3.1 VOC flux variation during the campaign**

During the three 9 day measurement periods in April, May and June Overall-22 compounds with a flux were found, of which 16 were identified by their elemental compositions (Table 1). Five compounds,  $C_4H_3O_4^+$  (formaldehyde),  $C_4H_4O_4N_4^+$  (formamide),  $C_4H_7O_4^+$  (crotonaldehyde) and two unidentified peaks with the masses of 84.9500 amu-Da and 118.9456 amuDa, had a negative net flux, each contributed around 1% or less to the total net flux (Fig. 3). As expected the average net flux increased from April-snowmelt (0.6624 nmol m-2 s-1) to Maythe start of the growing season (1.2207 nmol m-2 s-1) and June-summer (32.0075 nmol m-2 s-1). The compounds with detectable fluxes increased from three in during the snowmelt April-to 12 in Mayat the start of the growing season and 19 in June-summer. Over 8075% of the net flux comprised of methanol, acetone and7 monoterpenes-and butene. Of those threefour main compounds, acetone and monoterpenes had similar emission patterns (based on the total net flux) over the measurement period, while methanol had no detectable flux variation between day and night, whereas the periods in May and June showed a clear dependence on the temperature. The maximum emission was detected between 14:00 and 16:00 (Fig. 4); this is in agreement with the fact that VOC synthesis is driven by temperature and light (Ghirardo et al., 2010; Taipale et al., 2011), while potential evaporation from storage pools is primarily driven by temperature alone (Guenther et al., 1993). The maximum temperatures were typically measured induring mid-afternoon, when the light availability has still not yet decreased to a

great extent when compared to the light conditions around noon. Figure 5 shows that the highest emissions of monoterpenes and isoprene coincided with the highest temperatures. This can be seen in Fig. 5 where the highest emissions of the globally most emitted VOCs (excluding methane), the group of monoterpenes and isoprene, are during the hottest period of the campaign. Furthermore, the high monoterpene emissions during low PAR (<200  $\mu$ mol m-2 s-1; grey data points in Fig. 5) conditions can be explained by pool emissions, whereas the de novo isoprene emissions during this time were low. Unlike the maximum emission time, that-which was similar for all months, the minimum net flux was between 20:00 and 21:00 in-during the snowmeltApril, 03:00 and 04:00 inat the start of the growing season May-and 01:00 and 02:00 in Junesummer. Table 1 shows all the compounds with detectable flux for the three months and their 24\_h average emission and deposition.

**3.1.1 Low emissions during snow melt**

As expected the total emission  $(0.6825 \text{ nmol m}^2 \text{ s}^{-1})$  was smallest during the measurement period in April in April (compared to May-the start of the growing season and summerand June). The snow melted during this period and the average temperature and photosynthetically active radiation (PAR) were at their lowest, 4.4°C and 268 µmol m-2 s-1, respectively. The total deposition (-0.01 nmol m-2 s-1) was also weakest. Butene (C4H5+) dominated the emissions, comprising over 60% of the net flux. C4H9+ had the highest emissions between 14:00 and 15:00 with 0.82 nmol m-2 s-1 and the lowest between 21:00 and 22:00 with 0.08 nmol m-2 s-1. Acetone contributed with almost-over 250% to the emission peaked with 0.40 nmol m-2 s-1. The heaviest measured compounds with detectable flux was-were the group of monoterpenes, which contributed 1748% to the total emission and had the highest emissions between 14:00 and 15:00 and 15:00 with 0.24 nmol m-2 s-1. The lowest emissions of 0.06 nmol m-2 s-1 were measured during morning between 06:00 and 07:00. The anthropogenic emissions of butene & butanol dominated by a factor of 1.7 over the biogenic emissions. C4H9+ had the highest emission m-2 s-1 and the lowest between 21:00 and 22:00 with 0.82 nmol m-2 s-1 were measured during morning between 06:00 and 07:00. The anthropogenic emissions of 0.06 nmol m-2 s-1 and the lowest between 21:00 and 22:00 with 0.82 nmol m-2 s-1 and the lowest between 21:00 and 22:00 with 0.82 nmol m-2 s-1 had the highest emissions between 14:00 and 15:00 with 0.82 nmol m-2 s-1 had the highest emissions between 21:00 and 22:00 with 0.82 nmol m-2 s-1 had the highest emissions between 21:00 and 22:00 with 0.82 nmol m-2 s-1 and the lowest between 21:00 and 22:00 with 0.88 nmol m-2 s-1.

**3.1.2 Increase of emissions at start of growing season**

The total emission during the start of the growing season in May was 1.24 nmol m-2 s-1 and the total deposition was more than 10% of the emission, -0.17 nmol m-2 s-1. The night temperatures during this period were above zero and the sun warmed late afternoons to around 20°C. This led to a mean temperature of 11.4°C and the mean PAR was 301  $\mu$ mol m-2 s-1. Methanol dominated the emissions with 40% contribution to the total emission. The diurnal maximum of methanol was in the late afternoon with 2.11 nmol m-2 s-1. These results agree with other studies, which showed that methanol is released in plant growth (e.g. Galbally and Kirstine, 2002). On the other hand, methanol showed also the highest deposition, comprising almost 50% of the total deposition. The midday deposition of -0.47 nmol m-2 s-1 between 11:00 and 12:00 can be explained by rain during or right before this time window, which happened twice during the measurements in the May period. The water-soluble methanol was suspected to be dry deposited on the wet surfaces in the forest, (Laffineur et al., 2012; Wohlfahrt et al., 2015; Schallhart et al., 2016).

[revised manuscript text omitted]

**3.1.3 Maximum emissions during summer**

During the first weeks of June the highest average temperature and PAR were measured, with  $17.2^{\circ}$ C and  $466 \,\mu$ mol m-2 s-1, respectively. As temperature and light are the drivers of biogenic emissions (Guenther et al., 2012), the highest 24 h

total emission of the campaign, 2.87 nmol m-2 s-1, was recorded during this time (Table 1). In Fig. 4 the maximum diurnal 1 h net flux is shown at 14:30 with 9.10 nmol m-2 s-1. Similar as in the period in May, methanol, the group of monoterpenes and acetone were the most emitted compounds and the emissions of the ten most emitted compounds in summer all increased, if compared to the measurement period in May (Table 1).

In the summer period methanol was the most emitted compound, comprising 42% of the total emission and 17% of the deposition. The methanol flux was highest between 15:00 and 16:00 with 4.56 nmol m-2 s-1, while between 03:00 and 04:00 it was deposited (-0.26 nmol m-2 s-1). Growing leaf biomass is expected to release methanol due to cell wall demethylation (Galbally and Kirstine, 2002, Hüve et al., 2007, Aalto et al., 2014). The increase of the 24 h methanol emissions from undetectable during the snowmelt to about 0.5 nmol m-2 s-1 during the start of the growing season 
[revised manuscript text omitted]
 flux is almost a factor of 4 higher than in boreal forest in Hyytiälä. However, it should be noted that the day length in summer is much longer in Hyytiälä (62° N) when compared to the Ponderosa pine forest in Colorado (39° N) and, therefore, elevated emissions last longer than 8 h.

The net carbon flux of the VOCs during the campaign in Hyytiälä was \$7.0425 nmol C m-2 s-1 (Fig. 7). The group of monoterpenes was the highest emitter of carbon with 5459% of the net carbon exchange, followed by methanol (123%), acetone (1 $\pm2\%$ )s, butene (10%) and isoprene (5%) and acetaldehyde (3%). Acetaldehyde, tT he C3H5+ fragment and toluene & p-cymene contributed 2% respectively, while acetic acid and the sum of the remaining compounds both

contributed 1%. Compared to the CO2 net ecosystem exchange (NEE) of -4266 nmol C m-2 s-1, the carbon released as VOC represents less than 0.2% of the NEE of the corresponding period. In Brilli et al. (2015) the VOCs, with 6.36 nmol C m-2 s-1, represent 0.8% of the carbon exchange and in Schallhart et al. (2016) VOCs had a carbon flux of 41.8 nmol C m-2 s-1, which corresponded to 1.7% of the NEE. Juráň et al. (2017), measured EC exchange of VOCs in a *Picea abies* (Norway spruce) forest in the Czech Republic. In their study the ratio between the carbon released from VOCs and the NEE was 0.3% during the five days of data in July. Other ratios from vDEC studies using PTR-Quad are 0.7% above an oil palm plantation (*E. guineensis* × *E. oleifera* hybrid) in Malaysia (Misztal et al., 2011), 0.16% and 0.24% above two *P. halepensis* (Aleppo pine) forests in Israel (Seko et al., 2017).

There was an order of magnitude difference between the proportion of net assimilated carbon released as VOCs between a Mediterranean oak-hornbeam forest (Schallhart et al., 2016) and a boreal evergreen forest. Also, a middle European poplar plantation (Brilli et al., 2015) clearly released a higher proportion of the assimilated carbon as VOCs than a boreal site in this study. The spruce forest in central Europe came closest to the results in Hyytiälä. These findings strongly imply that there are significant differences between the ecosystems in how they allocate carbon to VOCs, however the reasons for that are rather related to light and thermal conditions, species selection and age structure and soil properties than to the efficiency of an ecosystem to produce VOCs as an indefinable concept. In boreal ecosystems with relatively northern locations, the majority of carbon assimilation is concentrated within a couple of months around mid-summer with very short nights. The structure of forest and the tree species are effective in utilizing the high light availability during the summer months, whereas in more southern locations the light availability and thermal conditions allow more even carbon assimilation throughout a considerably longer period. This partly explains the lower proportion of C released as VOCs determined in this study, when compared to those measured in more southern locations. Additionally, the constitutive emission capacities of boreal evergreen species are known to be low when compared to deciduous species more common in central and southern Europe (Rinne et al., 2009; Ghirardo et al., 2010).

**3.3 Comparison between PTR-TOF and PTR-Quad measurements**

Generally, the EC method detected more masses than the SLP measurements, which can be explained by the preselection of the masses to be measured by the PTR-Quad and its low duty cycle. The comparison between PTR-TOF using the EC method and the PTR-Quad using the surface layer profile method (SLP, see Rantala et al., 2015) between April and June revealed 12 more compounds with exchange using the EC method (see Table 1, 2 and 3). The PTR-TOF measures all VOCs in a certain mass range, while the PTR-Quad needs a preselection of masses, limiting the number of recorded compounds. However, the classification if a compound has a measurable flux is different between the two methods (Sect. 2.3 and Rantala et al., 2015). -Even though the EC method found almost twice the number of compounds with exchange, the total exchange was in on the same order of magnitude for both methods. The discrepancy of between the results of the two measurements are is mainly due to instrument and method differences. The horizontal distance between the inlets of these two instruments was just 25 m and the PTR-Quad was measuring from 13 m higher (calculated height 36 m) than the PTR-TOF, leading to a larger footprint area for the SLP method. This can be seen in Fig. 8, where the daytime (07:00 to 19:00) footprint calculated according to Kormann and Meixner, (2002) for the SLP and EC method are presented. Even though the nominal measurement height of the SLP was 36 m, the shown footprint is calculated for 33 m, as this was the closest height with horizontal wind measurements.

The main compounds (methanol, acetone and monoterpenes) were detected with by both methods.  $C_4H_9^+$  (57.0699 Damu) could not be measured by the PTR-Quad, as the unit mass resolution of the instrument is unable to separate the signal

from the water cluster isotope  $H_7O_2^{18}O_4^+$  (57.0432 Damu). On the other hand, the SLP measurements observed a flux of ethanol+formic acid which was not detected by the PTR-TOF. Interestingly, formic acid fluxes have also been observed at SMEAR II using the EC method with an Iodide-Adduct High-Resolution Time-of-Flight Chemical Ionization Mass Spectrometer in 2014 (Schobesberger et al., 2016). Other VOC fluxes, which were not detected by the PTR-TOF were methyl ethyl ketone (73 Damu), a fragment of the green leave volatiles (83 Damu) and MBO (87 Damu). The formaldehyde fluxes were not included in the comparison, as the PTR-TOF detected them just during the first 8 days in June, during which the PTR-Quad had technical problems resulting in less than ten overlapping data points from the two instruments.

Overall, the flux values used for the comparison were small, as the compared data included values from April and May, and unfortunately the SLP measurements were not working during the warm, i.e. high flux, period in the beginning of June. Thus, the comparison was done using flux values that were mostly close to the detection limits of the EC and the SLP setups. The uncertainties of the turbulence measurements and noise of the VOC measurements, together with the different measurement setups, methods and footprints was seen in the scatter of the compared data, which affected correlation, fitting parameters and their uncertainties (Table 3).

TheHowever, the magnitude of the studied fluxes as well as their diurnal patterns were comparable for methanol, acetone, isoprene and monoterpenes (Table 3 and Fig. 98). The monoterpeneand methanol fluxes measured by the two methods had showed also athe highest -good correlation, which was barely above 0.6-with each other (Fig. 109). The fitted slopes of the scatter plots for acetone, isoprene and the monoterpenes were far from unity, as best R2 values were calculated when using high intercepts. 
[revised manuscript text omitted]

| mass
[ <del>amuDa]</del> | elemental composition      | possible compound                       | Aprilsnowmelt    |       | Maystart of growing
season |       | Junesummer                             |       |
|------------------------------------|----------------------------|-----------------------------------------|------------------|-------|-------------------------------|-------|----------------------------------------|-------|
|                                    | -                          |                                         | E                | D     | E                             | D     | Е                                      | D     |
| 33.0335                            | $C_{4}H_{5}O_{4}^{+}$      | methanol                                |                  |       | <del>35.3</del> 39.6   | 49.1* | <del>38.742.</del>              | 16.5* |
|                                    |                            |                                         |                  |       |                               |       | 1                               |       |
| 137.1325                           | $C_{10}H_{17}^+$           | monoterpenes                            | 47.7 17.6 | 0     | 2 <del>3.6.02</del>    | 0     | <del>1921.2</del>               | 0     |
|                                    |                            |                                         |                  |       |                               |       | 0                               |       |
| 59.0491                            | $C_3H_7O_{\clubsuit^+}$    | acetone                                 | 52.3 19.3 | 100*  | 1 <del>6.8</del> 8.9   | 0     |  <del>13</del>14 . <del>1</del> | <1*   |
|                                    |                            |                                         |                  |       |                               |       | 3                               |       |
| 57.0699                            | $C_4H_9^+$                 | butene & butanol             | 170.5 63. | 0     | 1 <del>0.8</del> 2.1          | 0     | 8.2 9                           | 0     |
|                                    |                            |                                         | θ                |       |                               |       |                                        |       |
| 45.0335                            | $C_2H_5O_4{}^+$            | acetaldehyde                            |                  |       |                               |       | 5.4 5                           | 4.0*  |
| 69.0699                            | $C_5H_9^+$                 | isoprene                                |                  |       | 2. <del>2</del> 4             | <1*   | 4. <del>2</del>6                | 1.2*  |
| 61.0284                            | $C_{2}H_{5}O_{2}^{+}$      | acetic acid                             |                  |       |                               |       | 2.4 6                           | 4.1*  |
| 43.0178                            | $C_2H_3O_4{}^+$            | fragment                                |                  |       |                               |       | 2. <del>2</del> 4                      | 7.7*  |
| 41.0386                            | $C_3H_5^+$                 | fragment                                |                  |       | 2. <del>6</del>9       | 3.7*  | 2. <del>3</del>5                | <1*   |
| 31.0178                            | $C_{1}H_{3}O_{1}^{+}$      | formaldehyde                            |                  |       |                               |       | <1                                     | 41.4  |
| 60.0471                            | 60.0471                    | unknown                                 |                  |       | 3.4 8                  | 6.8*  | 1. <del>2</del> 3                      | 9.0*  |
| 93.0699                            | $C_7H_9^+$                 | toluene & p-cymene           |                  |       | 2.4 6                  | <1*   | <1                                     | 1.8   |
| 69.0352                            | 69.0352                    | unknown                                 |                  |       |                               |       | <1                                     | <1*   |
| 67.0542                            | $C_5H_7^+$                 | cyclopentadiene                         |                  |       |                               |       | <1                                     | <1*   |
| 70.0696                            | 70.0696                    | unknown                                 |                  |       | <1                            | 2.6   | <1                                     | 1.7*  |
| 99.0201                            | 99.0201                    | unknown                                 |                  |       |                               |       | <1                                     | 3.6*  |
| 84.9500                            | 84.9500                    | unknown                                 |                  |       |                               |       | <1*                                    | 4.8   |
| 95.0491                            | $C_6H_7O_{\textbf{L}}^+$   | phenol                                  |                  |       |                               |       | <1                                     | 1.1*  |
| 135.1168                           | $C_{10}H_{15}^+$           | p-cymene                                |                  |       |                               |       | <1*                                    | <1*   |
| 46.0287                            | $C_{1}H_{4}O_{1}N_{1}^{+}$ | formamide                               |                  |       | 4 2.91                 | 23.9  |                                        |       |
| 118.9456                           | 118.9456                   | unknown                                 |                  |       |  ≤<del><1</del>1 *  | 9.0   |                                        |       |
| 71.0491                            | $C_4H_7O_{\clubsuit}^+$    | MVK /& MACR                  |

---

## Author Response (AR1)

We would like to thank the reviewers for their detailed and helpful comments and suggestions, which improved the manuscript. The comments of the reviewers are in black, while our replies are marked in red. The revised manuscript is attached after the comments and answers to all reviewers.

**Anonymous Referee #1**

The paper by Schallhart et al. describes NMVOC flux measurements above a forested site in Finland using two PTR instruments. They calculate fluxes based on the direct eddy covariance and the indirect gradient method.

1) It is interesting that a much smaller number of actively exchanged VOCs were observed during this study than by Park et al., 2013, using similar methods for analyzing the entire mass spectrum. Is there an explanation? Was it a problem of the analysis (e.g. shifting time synchronization) or are Boreal forest generally just low (or not very active) BVOC emitting ecosystems? It appears that the measured net-fluxes are generally quite low when comparing with similar measurements from temperate and tropical forests.

A good question, we do not suspect the analysis being the cause of the discrepancy in the number of actively exchanged VOCs. The analysis was same as in Schallhart et al., 2016, where VOC fluxes with constant lag time (2.6 s) were measured above a very active (24 h net flux: 9.78 nmol m$^{-2}$ s$^{-1}$) oak forest. Still in the oak forest, only 29 compounds were actively exchanged. Schallhart et al. (2016) also gives a more detailed explanation of automated method and compares this method with a manual method.

Therefore, the reason in the huge difference between 494 (Park et al., 2013), 29 (Schallhart et al., 2016) and 17 (for 21 days in June; this study) exchanged compounds must be primarily caused by the behavior of the orange yard, which seems to be a unique ecosystem. The measurement setup, e.g. shorter inlet lines as used by Park et al. (2013), can reduce the losses for heavier, stickier and/or more reactive VOCs, but cannot explain an order of magnitude difference in number of exchanged compounds. Additionally, the meteorological circumstances (PAR, T) can lead to stress responses in plants and higher photochemistry could lead to additional oxidation products.

Overall, a comparison between different ecosystems is very challenging. The flux of individual compounds are similar as in previous boreal forest measurements indicating that the measurements of ecosystem fluxes worked well.

If the oak forest and the boreal forest are compared, the difference in number of actively exchanged compounds should be mainly caused by the lower emissions in the boreal forest and fewer compounds are released in detectable amounts. The shift in lag time should have a minor effect (after the described corrections).

2) No flux footprint analysis is presented. Why? It would seem important for the interpretation of the results, especially for a site which is largely comprised of a managed forest (with clearings). I also see there is a nearby lake. Knowing the footprint might also be a very relevant issue for the comparison between the two flux methods. The analysis would also seem to be crucial for the interpretation of butanol contamination from aerosol measurements (section 3.4). The plotted wind rose suggests that a large proportion of the wind-direction is from the S-W sector, where the buildings of the research station are situated.

Thank you for your comment, we agree that footprints would be important for interpreting the results. Therefore, we added average cumulative footprints (50% and 80%) for both the EC and the SLP measurements. The average footprints were calculated from the measurement period using daytime

values (07:00 – 19:00). During night, the flux footprint would be generally larger due to stronger stable conditions, but the VOC emissions are, on the other hand, usually low.

Cumulative footprints were estimated using an analytical function by Kormann and Meixner (2001). The algorithm uses horizontal wind speed, Obukhov length, standard deviation of a lateral wind, roughness length (estimated to be ca. 1.5 m), measurement height and zero displacement height (we used a value 13 m) to determine the footprint function. According to Horst (1999), footprints are similar for both the EC and the SLP methods if the measurement heights are equal. Therefore, we used the same algorithm also for the SLP fluxes.

Determined footprints have error sources and thus the results should be taken as estimates. The nominal measurement height of the SLP fluxes was 36 m (for unstable stratification). However, we used the height 33 m because horizontal wind measurements were available from that level. This underestimates slightly the footprints. On the other hand, Kormann & Meixner algorithm excludes an effect of roughness sublayer which leads to an underestimation of the turbulent transport and thus overestimates the cumulative footprints.

3) m/z 93: While this mass is commonly shown to be associated with toluene using PTR ionization, there is evidence that a fragment from cymene can exhibit some interference (e.g. 20-30%) above evergreen forests. Have the authors verified that this is not the case – e.g. by correlating fluxes or concentrations of the corresponding ions? I also wonder whether benzene fluxes were detectable, and whether benzene to toluene (concentrations and/or fluxes) ratios could indicate local pollution.

We agree that para-cymene can influence the signal and stated it in the end of Sect. 2.2. With the E/N used in this study, over 80% of the para-cymene signal should fragment to 93.0699 Da (Tani et al., 2003). Our intention was to name each signal depending on the mass where it was measured and discuss the possible compounds in Sect. 2.2. However, we realize that this can lead to misinterpretations and therefore will rename all signals measured at mass 93.0699 from toluene to toluene & p-cymene.

The correlation between the p-cymene (135.1168 Da) and toluene & p-cymene (93.0699 Da) concentration was 0.79. Due to the small fluxes of p-cymene (135.1168 Da), the 30 min files (used for the correlation) were influenced by noise, leading to a correlation with toluene & p-cymene (93.0699 Da) beneath 0.3.

Benzene fluxes were not detected and the average toluene & p-cymene/ benzene concentration ratio in June was 1.26 (standard deviation: 0.51). . The toluene/benzene ratio from fresh anthropogenic plume is 4.25 (Warneke et al., 2007) and in the order of 2 for fresh traffic and the ratio goes down as the air mass is oxidized during transported from the source (Gelencsér et al., 2007) indicating that the source is not dominated by local anthropogenic or traffic sources.

4) Section 3.2. While being a valuable section, a number of studies are missing here – it would benefit the manuscript to conduct a more thorough literature review.

We thank the reviewer for this comment. The aim in this paper is to give an overview over the relative new and few already published studies, where no preselection of VOCs had to be made and, therefore, all VOCs with a flux above the detection limit (excluding instrument related signals) are reported. Thus the literature of VOC fluxes was focused on comparing EC fluxes measured with PTR-TOF that can scan the full range of VOCs emitted, instead of studies with a preselected set of studied compounds, typically 10-20 masses with the DEC fluxes using the PTR-Quad. Furthermore, we omitted publications which use less than four days of data (e.g. Ruuskanen et al., 2011) as fluxes would be prone to statistical variation and, therefore, be not representative of the ecosystem. Also studies which concentrate on specific VOCs are disregarded (e.g. Fares et al., 2013; Su et al., 2016).

Thus, as stated, we would like just to include EC fluxes (measured with PTR-TOF), as a comparison with DEC fluxes (measured by PTR-Quad) would be out of the scope of this study and would need, due to the large amount of studies, an individual review paper.

We added Juráň et al. (2017), where approximately five days of VOC fluxes were measured using EC with a PTR-TOF. However, we added the fraction of the assimilated carbon released from VOCs from Misztal et al. (2011) and Seco et al. (2017).

5) Table 2: 24 h flux average are chosen quite randomly and are often problematic, because most of the studies do not apply corrections to night time fluxes (e.g. storage and advection). A couple of clarifications would be important. Which (if any) corrections were applied to night time fluxes? Also why compare a 24h average with an 8h average – this does not make much sense. One should average the current dataset over the same period as each of the referenced datasets to be a meaningful comparison.

We agree that 24 h flux values are problematic, as the night time fluxes are more prone to errors. As suggested we added 8 h average flux values from this study, to compare to the values of Kaser et al. (2013) and clearly marked them as 8 h averages.

However, especially when measuring in northern countries, where emission are low but continue longer than in mid latitude areas, 8 h fluxes can lead to an underestimation. In high latitudes the sun sets only for a few hours in the night in summertime and temperature as well as light and temperature dependent emissions continue to midnight (maximum length of daytime light in this study: 19.5 h) and using midday 8 h flux averages would further decrease the cumulative emissions . We clarified in Sect. 2.3 that no additional corrections were applied to the measured fluxes (in addition to the high frequency attenuation correction).

6) Page 10 Line 32: So if you conclude that the SMEAR station generally suffers from widespread butanol contamination from aerosol instrumentation, why talk about a 'butene flux' in this paper?

Similarly as discussed in question 3, the initial idea was to always use the compound name of the measured mass and discuss the identification of the compound in Sect. 2.2. We agree that this can lead to misinterpretations and changed all butene references to butene & butanol. We would like to keep butene in the name, as a separation of the $C_4H_9^+$ signal into butene and butanol fragment is, with our measurements setup, not possible. However, we clarify at several points that the butene & butanol emissions are anthropogenic and refer to Sect. 3.4.

There are a large number of grammatical errors or confusing statements– I suggest copy-editing if this cannot be easily resolved by the authors. A small number of examples is listed below.

We thank the reviewer for this comment the manuscript was checked carefully for mistakes after all reviewer comments were implemented.

Page 5 Line 28: Overall 22 compounds with flux – change to "with a flux"
Page 5 Line 29: "C1" is an odd way to present a sum formula – I suggest to change it to "C" throughout the manuscript
Page 6 Line 4: measured in mid-afternoon – change to "during mid-afternoon"
Page 6 Line 6: methane is typically not considered a VOC – I suggest changing VOC to NMVOC throughout the manuscript as this is a more commonly used expression
We agree with the reviewer, however, we would like to stay with the term VOC. Therefore the sentence was rewritten and any reference to methane was omitted.

Page 6 Line 6: The entire sentence here is confusing – I suggest to rephrase it.
Page 6 line 9: "for all months"
Page 6 line 10: "detectable fluxes"
Page 6 line 20: The heaviest measured compounds with detectable flux was - > "were"
Page 9 line 4: using surface layer profile method - > using "THE" surface layer method
Page 9 line 7: "ON" the same order
Page 9 line 8 : between the two measurements ?
Was changed to: The discrepancy between the results of the two measurements is mainly […]
Page 9 Line 12: "by" both methods
Page 11: "air chemistry"
Table 1 and 2, Figure 3, 6, 7: change "C1, O1, N1, etc." in the elemental composition column to "C, O, N"
All above mentioned examples were corrected.

Note: The unit of mass was changed from amu to Da, to ensure that the reader knows that the mass units used are based on the $^{12}C$ atom.

[Figure]

**Figure Z1: Mass dependency of the transmission of the PTR-Quad (left) and scatter of the measured and calculated normalized sensitivity.**

**Anonymous Referee #3**

Schallhart and colleagues present VOC flux measurements at the SMEAR II station in Southern Finland. The measurements reported are interesting to the biogenic VOC flux community, however, some more work needs to be done to be able to accept this manuscript for publication. In addition, the language should be edited throughout the manuscript in order to improve sentence structure and thus text readability. Some issues that deserve attention are shown in the list below.

P1L13 Correct 'only a three' to 'only three'

changed

P1L1-2 This sentence needs some English editing

We rephrased the sentence to: "The VOC exchange followed a seasonal trend and the emissions increased from spring to summer"

P2L5 "seems" should be "seem"

changed

P3L37-39 rephrase

We rephrased the section.

P4L17-25 It is not clear to me what is the advantage of the second step, because the second and third steps seem almost the same thing. Please clarify the reason why those steps were taken.

We thank the reviewer for this comment. After the first step, all lag times are similar than the lag time of the monoterpenes. The second step is used to correct compound specific lag times, e.g. if certain compounds would have on average a shorter or longer lag time, e.g. due to interaction with RH or tube walls (similar to the response time). This correction is constant over all the measurement period and ensures that the next step (three) uses a +- 10 s time window around the maximum.
In step 3 every 30 min flux point individually is corrected. This corrects small fluctuations in the lag time, e.g. when the linear regression fit was not precise enough.
We added a sentence explaining step 2 to the revised manuscript.

P5L6 Can the authors list the values of average correction factors for day and night separately?

Yes, we added the average daytime value of 16% (between 09:00 and 17:00) and nighttime value 23% (between 20:00 and 04:00).

P5L19 why the period of 21 days from 04 to 24 May is shown when only a period of 9 days was used?

During May we had technical problems with the 3d-sonic anemometer, which stopped recording data during several occasions during the period of 04 May 2013 to 24 May 2013. Therefore, only 423 30 min files (equals 211.5 h, approximately 9 days of data) of fluxes could be calculated in this period. We clarified this section in the manuscript.

P6L25-26 confusing sentence

We rewrote the sentence and improved the readability.

P6L24-P7L34 Sections 3.1.2 and 3.1.3 should be rewritten to emphasize readability

We rewrote the mentioned sections and improved the readability.

P7L10 does this "highest total emission" refer to the maximum "average daytime emission"? Please clarify since it is difficult to discern from the (higher) 9.69 nmol/m2s shown on P7L13.

We changed the "highest total emission" to "highest 24 h total emission". We did not use "average daytime emission", as the mentioned value includes also nighttime emissions.

P7L36-P8L38 There are many other recent studies that, either with PTR-TOF or PTRQuad, reported VOC fluxes that would give a more complete view to your discussion. Some of these studies report what fraction of the assimilated carbon was released as VOCs. In addition, some studies report VOC fluxes from monoterpene-emitting conifer species, which may be more relevant to your study. Some examples:
- McKinney et al (2011) https://doi.org/10.5194/acp-11-4807-2011
- Misztal et al (2011) https://doi.org/10.5194/acp-11-8995-2011
- Kalogridis et al (2014) https://doi.org/10.5194/acp-14-10085-2014
- Seco et al (2015) https://doi.org/10.1111/gcb.12980
- Juráˇn et al (2017) https://doi.org/10.1016/j.agrformet.2016.10.005
- Seco et al (2017) https://doi.org/10.1016/j.agrformet.2017.02.007

We thank the reviewer for the suggested literature and also refer to Reviewer 1 Q4:
As stated we would like to include just EC fluxes (measured with PTR-TOF), as a comparison with DEC fluxes (measured by PTR-Quad) would be out of the scope of this study and would need, due to the amount of studies, an individual review paper.
The aim in this paper was to give an overview over the relative new and few already published studies, where no preselection of VOCs had to be made and, therefore, all VOCs with a flux above the detection limit (excluding instrument related signals) are reported.
Furthermore, we omitted publications which use less than four days of data (e.g. Ruuskanen et al., 2011) as fluxes would be prone to statistical variation and, therefore, be not representative of the ecosystem. Also studies which concentrate on specific VOCs are disregarded (e.g. Fares et al., 2013; Su et al., 2016).
We added Juráň et al. (2017), where approximately five days of VOC fluxes were measured using EC with a PTR-TOF. However, as 24 h flux values were not stated for all measured compounds, the study was not added in Table 2. We also we added the fraction of the assimilated carbon released from VOCs from Misztal et al. (2011) and Seco et al. (2017).
McKinney et al. (2011) and Kalogridis et al. (2014) were not added, as no VOC/NEE carbon fraction was stated. Seco et al. (2015) was not used, as just the daytime fraction is stated.

P9L5 Should the text "12 more compounds" be interpreted as 12 compounds exhibiting SLP fluxes not found with the TOF-only EC analysis?

No, the opposite, there were 12 more compounds measured with the EC method that were not found with the SLP method. We clarified this in the text.

P9L22 Figure 8 shows indeed that fluxes for monoterpenes, isoprene and acetone were comparable. However, Table 3 lists the slope of the fit between the fluxes calculated with both methods and the slopes for all compounds (including monoterpenes) except acetone are far away from 1. The authors should further discuss this discrepancy, in addition to what is already discussed in P10L4-10. Also please rephrase the last sentence because from the text it seems that monoterpene and methanol were correlated with each other, when Fig 9 shows that what was correlated were the fluxes calculated with EC and SLP.

The mentioned major discrepancy is caused by the high intercept of the fitting. In the case of the monoterpenes, the intercept is around 50% of the mean flux, explaining the flat slope of 0.5. When we forced the offset to be zero, the slope increase to 0.7. The still existing discrepancy from unity can be partly explained by the outliers (Fig. 9), which are far off the 1:1 line and thereby decrease the slope (with their high influence on the explained sum of squares).
A similar case is the slope of isoprene. Here, the offset is about 30% of the mean flux and again, by forcing a zero intercept the slope increased. Furthermore, a close look at Fig. 8 shows, that there is clearly a smaller flux measured by the EC method, when compared with the SLP method and thereby the slope should be less than one.
Acetone has the highest intercept compared to the mean flux ($> 60\%$) and also higher uncertainty of the slope ($0.276 \pm 0.191$) with respect to the compounds mentioned above.
When the offset was forced to be zero, the $R^2$ values decreased for all compounds.
Overall, the discrepancy from unity can be explained by the scatter of the measured fluxes, which is shown in the uncertainties of the fitting parameters. The scatter is caused, as discussed in the paper, by variation in turbulence measurements, noise of the VOC measurements, different inlet heights (36 m, 23 m), methods (SLP, EC), instruments (PTR-Quad, PTR-TOF) and footprints (see revised manuscript Fig. 8). The comparison also lacked high flux periods, as can be seen when comparing the values of the monoterpene EC flux in Fig 8 (right side; 3 values above 1 nmol m$^{-2}$ s$^{-1}$) and Fig. 4 (bottom panel), where in the 9 day period already 44 values were above 1 nmol m$^{-2}$ s$^{-1}$ (maximum emission 2.44 nmol m$^{-2}$ s$^{-1}$).
We added a discussion about the discrepancy of the fitted slopes from unity to the manuscript.

P10 section 3.4 the contamination from butanol used in CPCs is a major problem for the attempt to report butene fluxes. Since the authors have concluded that most of the signal of CH4H9+ is from butanol, the results and discussion section relating to butene should be omitted. I would recommend still having part of the manuscript dedicated to explaining what happened with butanol so other researchers can keep this contamination in mind for future experiments. Also, have the authors quantified the capacity of the forest (e.g. the fluxes measured on the eastern side of the forest) to take up butanol?

See also the answer to Reviewer 2 Q8:
We agree with the reviewer, that the anthropogenic nature of the butene & butanol flux is lost, if just parts of the manuscript are read. Therefore we will omit the artificial butene & butanol flux in the total emission (Table 1 and 2), mentioned net fluxes and in Fig. 7. Still, the individual values will be presented in Table 1 and 2 as well as in Figs. 3, 4 and 6 as we would like to inform the reader that these fluxes do exist. All values connected with butene & butanol were labelled as anthropogenic to clarify that the flux does not come from the same biogenic source as other VOCs.
In the Fig. 10 left panel a small deposition of butene & butanol has been found with -0.04 nmol m$^{-2}$ s$^{-1}$ (0°-30° sector, NNE) and $< 0.01$ nmol m$^{-2}$ s$^{-1}$ (30°-60°, NE) sector. However, if this is active uptake of the forest or dry deposition we cannot answer with our measurement setup.

P10L28 How many compounds showed fluxes? Here and in section 3.2 the authors say 25, but in section 3.1 they say 22. Please be consistent.

We thank the reviewer for the comment and clarified the text. Overall, 25 compounds with fluxes have been discovered, 17 during the 21 day measurements in June, and 22 during the 3 periods in April, May and June. We rephrased the sentence in Sect. 3.1.

P11L6-8 A good reference for this final sentence is Rinne et al 2016 https://doi.org/10.1016/j.atmosenv.2016.02.005

We added the reference.

Fig 10 caption: "butanol using aerosol instruments" is confusing

Was changed to "[…] aerosol instruments, which use butanol."

Note: The unit of mass was changed from amu to Da, to ensure that the reader knows that the mass units used are based on the $^{12}C$ atom.

[revised manuscript text omitted]

The flux underestimation caused by high frequency attenuation was estimated using a parametrization described by Horst (1997). The method uses a system response time and information about stability and horizontal wind speed for estimating the attenuation. The system response time was determined to be around 1.2 s for monoterpenes and the same response time was also used for all the other compounds. ~~However, the response time of e.g. water soluble compounds might be larger due to desorption and absorption processes on the tube walls, as a function of relative humidity and sampling line aging (Mammarella et al, 2009; Nordbo et al., 2013 and 2014). This potentially causes errors of a few percentage on the flux values. However, we expect the effect to be reduced, because a heated sampling tube was used. The low-frequency corrections that are based on theoretical transfer function shapes (e.g. Rannik and Vesala, 1999), were not applied in this study.~~

One should note that the determined response time describes the flux attenuation of the whole measurement setup, including the tubing, a horizontal separation between the inlet and the anemometer and the instrument itself.  In this case, the average attenuation factor was 18%. On average, the correction factor was smaller during the day (16%; 09:00 to 17:00) and larger at night (23%; 20:00 to 04:00).

No additional corrections were applied to the measured fluxes (e.g. storage correction).

**2.4 Flux quality criteria**

The measured fluxes were filtered by three quality criteria, to reduce the systematic uncertainty and ensure their representativeness:

First, the data were flagged if the tilt angle, resulting from the coordinate rotation of sonic anemometer wind velocity components (Kaimal and Finnigan, 1994), was more than 5°, which was the case for 11.9% of the data. Second, all 30 min records with a friction velocity less than 0.2 m s$^{-1}$ were flagged. Following this, 11.2% of the data  were flagged. Finally, the flux steady-state test was applied according to Foken and Wichura (1996). All flagged flux values were removed from further analysis. The rejection rate between April and June was 34.1%, 35.1% and 30.5% for acetone, butene & butanol and the monoterpenes, respectively. For the monoterpenes, the rejection rates for the measurement periods in April, May and June were 19.1%, 17.6% and 30% (daytime) and 25.6%, 24.0% and 43.7% (nighttime).

**2.5 Flux selection**

The temporal behavior of the VOC exchange was investigated by measuring periods in April, May and June 2013. These periods give insight to the VOC exchange during snowmelt, start of the growing season and summer.  Because of technical problems with the anemometer, which stopped recording data on several occasions during the start of the growing season, only 423 × 30 min files (~9 days of data) of VOC fluxes could be calculated in the period from 04 May 2013 to 24 May 2013. The standard deviation of noise in the averaged, absolute CCFs ($\sigma_{noise}$) determines the exchange threshold and is directly dependent on the amount of data. Therefore, the same amount of data (423 × 30 min files) was selected to represent each  period from snowmelt to summer and make a comparison between those periods possible. For all three periods, the absolute

mean of the CCFs between 10:00 and 16:00 (UTC+2) was used to find compounds with statistically significant flux (Park et al., 2013; Schallhart et al., 2016). For the snowmelt, the measurements were from 14 April 2013 to 24 April 2013 and for  summer the hottest period was selected, from 01 June 2013 to 12 June 2013.

**3. Results and Discussion**

The PTR-TOF measures the mass of the VOCs in the ambient sample, which can be used to calculate the elemental composition. Therefore, no structural information of the measured molecules is known and the identification of compounds relies on literature and gas chromatograph measurements. Major masses affected by fragmentation and compounds with high uncertainties are discussed in the following.

The mass 69.0699 Da with the elemental composition $C_5H_9^+$ was called isoprene & MBO, as 2-methyl-3-buten-2-ol (MBO) fragmented to this mass and had a substantial influence on the signal (e.g. Kaser et al., 2013b). Similarly, the mass 93.0699 Da with the elemental composition $C_7H_9^+$ was called toluene & p-cymene, as p-cymene fragments were suspected to affect the signal (Tani et al., 2003). Formaldehyde has only a slightly higher proton affinity compared with the primary ion and therefore back reactions, which are humidity dependent, from protonated formaldehyde to water occur (de Gouw and Warneke, 2006; Inomata et al., 2008; Vlasenko et al., 2010). This may have led to an artificial flux, which was caused by water vapor fluctuations. Therefore, the formaldehyde fluxes are very uncertain. The signal at mass 57.0699 with a protonated composition of $C_4H_9^+$ was called butene & butanol, as butanol was suspected to have substantial influence on the signal (see Sect. 3.4). Furthermore, as the butene & butanol fluxes were expected to be caused by the aerosol instrumentation, they were disregarded in all reported net fluxes and total emissions. The monoterpenes ($C_{10}H_{17}^+$) were measured at mass 137.1325 Da only. The monoterpene fragment at mass 81.0699 Da ($C_6H_9^+$) was identified by its Pearson correlation of 0.99 (30 min integrated data) with the signal at 137.1325 Da and was disregarded from further analysis.

**3.1 VOC flux variation during the campaign**

During the three 9 day measurement periods in April, May and June  22 compounds with a flux were found, of which 16 were identified by their elemental compositions (Table 1). Five compounds, $C_4H_3O_4^+$ (formaldehyde), $C_4H_4O_4N_4^+$ (formamide), $C_4H_7O_4^+$ (crotonaldehyde) and two unidentified peaks with the masses of 84.95  Da and 118.9456  Da, had a negative net flux, each contributed around 1% or less to the total net flux (Fig. 3). As expected the average net flux increased from  snowmelt (0.24 nmol m$^{-2}$ s$^{-1}$) to the start of the growing season (1.207 nmol m$^{-2}$ s$^{-1}$) and  summer (2.075 nmol m$^{-2}$ s$^{-1}$). The compounds with detectable flux increased from three  during the snowmelt  to 12 at the start of the growing season and 19 in summer. Over 75% of the net flux comprised of methanol, acetone and monoterpenes . Of those four main compounds, acetone and monoterpenes had similar emission patterns (based on the total net flux) over the measurement period, while methanol had no detectable flux in April. The development of the diurnal cycle can also be seen in Fig. 4, as the measurements in April had a minor flux variation between day and night, whereas the periods in May and June showed a clear dependence on the temperature. The maximum emission was detected between 14:00 and 16:00 (Fig. 4); this is in agreement with the fact that VOC synthesis is driven by temperature and light (Ghirardo et al., 2010; Taipale et al., 2011), while potential evaporation from storage pools is primarily driven by temperature alone (Guenther et al., 1993). The maximum temperatures were typically measured during mid-afternoon, when the light availability has still not yet decreased to a

great extent when compared to the light conditions around noon. Figure 5 shows that the highest emissions of monoterpenes and isoprene coincided with the highest temperatures.  Furthermore, the high monoterpene emissions during low PAR (<200 µmol m$^{-2}$ s$^{-1}$; grey data points in Fig. 5) conditions can be explained by pool emissions, whereas the de novo isoprene emissions during this time were low. Unlike the maximum emission time,  which was similar for all month, the minimum net flux was between 20:00 and 21:00  during the snowmelt, 03:00 and 04:00 at the start of the growing season  and 01:00 and 02:00 in summer. Table 1 shows all the compounds with detectable flux for the three months and their 24 h average emission and deposition.

**3.1.1 Low emissions during snow melt**

As expected the total emission (0.25 nmol m$^{-2}$ s$^{-1}$) was smallest during the measurement period in April  (compared to  the start of the growing season and summer). The snow melted during this period and the average temperature and photosynthetically active radiation (PAR) were at their lowest, 4.4°C and 268 µmol m$^{-2}$ s$^{-1}$, respectively. The total deposition (-0.01 nmol m$^{-2}$ s$^{-1}$) was also weakest.  Acetone contributed with  over 20% to the emissions and was the only compound  during the snowmelt for which a diurnal deposition was detected. Between 22:00 and 23:00 the measured flux reached a minimum of -0.12 nmol m$^{-2}$ s$^{-1}$, whereas between 13:00 and 14:00 the emission peaked with 0.40 nmol m$^{-2}$ s$^{-1}$. The heaviest measured compounds with detectable flux  were the group of monoterpenes, which contributed 48% to the total emission and had the highest emissions between 14:00 and 15:00 with 0.24 nmol m$^{-2}$ s$^{-1}$. The lowest emissions of 0.06 nmol m$^{-2}$ s$^{-1}$ were measured during morning between 06:00 and 07:00. The anthropogenic emissions of butene & butanol dominated by a factor of 1.7 over the biogenic emissions. C$_4$H$_9$$^+$ had the highest emissions between 14:00 and 15:00 with 0.82 nmol m$^{-2}$ s$^{-1}$ and the lowest between 21:00 and 22:00 with 0.08 nmol m$^{-2}$ s$^{-1}$.

**3.1.2 Increase of emissions at start of growing season**

The total emission during the start of the growing season in May was 1.24 nmol m$^{-2}$ s$^{-1}$ and the total deposition was more than 10% of the emission, -0.17 nmol m$^{-2}$ s$^{-1}$. The night temperatures during this period were above zero and the sun warmed late afternoons to around 20°C. This led to a mean temperature of 11.4°C and the mean PAR was 301 µmol m$^{-2}$ s$^{-1}$. Methanol dominated the emissions with 40% contribution to the total emission. The diurnal maximum of methanol was in the late afternoon with 2.11 nmol m$^{-2}$ s$^{-1}$. These results agree with other studies, which showed that methanol is released in plant growth (e.g. Galbally and Kirstine, 2002). On the other hand, methanol showed also the highest deposition, comprising almost 50% of the total deposition. The midday deposition of -0.47 nmol m$^{-2}$ s$^{-1}$ between 11:00 and 12:00 can be explained by rain during or right before this time window, which happened twice during the measurements in the May period. The water-soluble methanol was suspected to be dry deposited on the wet surfaces in the forest, (Laffineur et al., 2012; Wohlfahrt et al., 2015; Schallhart et al., 2016).

The monoterpenes were the second most emitted compound group and contributed 26% to the total emission. Their maximum emission was 0.76 nmol m$^{-2}$ s$^{-1}$ between 15:00 and 16:00 and the minimum emission was 0.13 nmol m$^{-2}$ s$^{-1}$ between 3:00 and 4:00. Recently, Aalto et al. (2014, 2015) have shown that Scots pine needles are a pronounced source

of monoterpenes in spring already before growth onset, and especially after bud break, when the formation of new biomass releases large amounts of terpenoids and other VOCs. The results of this study are in general consistent with those findings in terms of detected mean fluxes and diurnal patterns. Acetone contributed with 19% to the total emission and was the third most emitted compound. It had the maximum emission of 0.61 nmol m$^{-2}$ s$^{-1}$ between 10:00 and 11:00 and the minimum between 03:00 and 04:00 with 0.04 nmol m$^{-2}$ s$^{-1}$. Formamide passed the 3 $\sigma_{noise}$ criteria only in the May period, where it explained 2% of the total emission and 24% of the total deposition. The emissions were highest between 19:00 and 20:00 with 0.13 nmol m$^{-2}$ s$^{-1}$ and the deposition peaked between 12:00 and 13:00 with -0.20 nmol m$^{-2}$ s$^{-1}$.

The emissions of butene & butanol, which is discussed in Sect. 3.4, were not related to the start of the growing season. The flux of C$_4$H$_9^+$ decreased by almost two-thirds compared to the snowmelt period and would increase the total emission by 11%. The maximum flux of 0.29 nmol m$^{-2}$ s$^{-1}$ was between 15:00 and 16:00 and the minimum of 0.02 nmol m$^{-2}$ s$^{-1}$ between 22:00 and 23:00. However, during the start of the growing season, most of the emissions were biogenic.

~~Methanol is released in plant growth (e.g. Galbally and Kirstine, 2002), its emissions increased from being not detectable in April to a late afternoon maximum of 2.11 nmol m$^{-2}$ s$^{-1}$ in May. Compared to April's emissions the order of the major emitters reverses in addition to the increase of compounds with observable exchange. In May night temperatures were above zero, while the sun warmed late afternoons to around 20°C. The mean temperature was 11.4°C and the mean PAR was 301 µmol m$^{-2}$ s$^{-1}$, during the measurements in May. The total deposition was higher than in the other months and reached -0.13 nmol m$^{-2}$ s$^{-1}$, which is less than 10% of the total emission (1.37 nmol m$^{-2}$ s$^{-1}$). Methanol was the most emitted (35%) and deposited compound (49%) in May, with the highest deposition of -0.47 nmol m$^{-2}$ s$^{-1}$ between 11:00 and 12:00. This deposition can be explained by the occurrence of rain during or right before this time window, which happened twice during the 9 day measurement period. The water soluble methanol is suspected to be dry deposited on the wet surfaces in the forest, (Laffineur et al., 2012; Wohlfahrt et al., 2015; Schallhart et al., 2016). The monoterpenes were the second most emitted compound group and contributed 23% to the total emission. Their maximum emission was 0.76 nmol m$^{-2}$ s$^{-1}$ between 15:00 and 16:00 and the minimum emission was 0.13 nmol m$^{-2}$ s$^{-1}$ between 3:00 and 4:00. Recently, Aalto et al. (2014, 2015) have shown that Scots pine needles are a pronounced source of monoterpenes in spring already before growth onset, and especially after bud break, when the formation of new biomass releases large amounts of terpenoids and other VOCs. The results of this study are in general consistent with those findings in terms of detected mean fluxes and diurnal patterns. During the start of the growing season, acetone was the third most emitted compound. In May it had the maximum emission of 0.61 nmol m$^{-2}$ s$^{-1}$ between 10:00 and 11:00 and the minimum between 03:00 and 04:00 with 0.04 nmol m$^{-2}$ s$^{-1}$. The flux of C$_4$H$_9^+$ decreased by almost two-thirds compared to April comprising 11% of the total emission. The maximum flux of 0.29 nmol m$^{-2}$ s$^{-1}$ was between 15:00 and 16:00 and the minimum of 0.02 nmol m$^{-2}$ s$^{-1}$ between 22:00 and 23:00. Formamide passed the 3 $\sigma_{noise}$ criteria only in May, where it explained 2% of the total emission and 24% of the total deposition. The emissions were highest between 19:00 and 20:00 with 0.13 nmol m$^{-2}$ s$^{-1}$ and the deposition peaked between 12:00 and 13:00 with -0.20 nmol m$^{-2}$ s$^{-1}$.However, most of the emissions, were biogenic.~~

**3.1.3 Maximum emissions during summer**

During the first weeks of June the highest average temperature and PAR were measured, with 17.2°C and 466 µmol m$^{-2}$ s$^{-1}$, respectively. As temperature and light are the drivers of biogenic emissions (Guenther et al., 2012), the highest 24 h

total emission of the campaign, 2.87 nmol m$^{-2}$ s$^{-1}$, was recorded during this time (Table 1). In Fig. 4 the maximum diurnal 1 h net flux is shown at 14:30 with 9.10 nmol m$^{-2}$ s$^{-1}$. Similar as in the period in May, methanol, the group of monoterpenes and acetone were the most emitted compounds and the emissions of the ten most emitted compounds in summer all increased, if compared to the measurement period in May (Table 1).

[revised manuscript text omitted]

contributed 1%. Compared to the $CO_2$ net ecosystem exchange (NEE) of -4266 nmol C m$^{-2}$ s$^{-1}$, the carbon released as VOC represents less than 0.2% of the NEE of the corresponding period. In Brilli et al. (2015) the VOCs, with 6.36 nmol C m$^{-2}$ s$^{-1}$, represent 0.8% of the carbon exchange and in Schallhart et al. (2016) VOCs had a carbon flux of 41.8 nmol C m$^{-2}$ s$^{-1}$, which corresponded to 1.7% of the NEE. Juráň et al. (2017), measured EC exchange of VOCs in a *Picea abies* (Norway spruce) forest in the Czech Republic. In their study the ratio between the carbon released from VOCs and the NEE was 0.3% during the five days of data in July. Other ratios from vDEC studies using PTR-Quad are 0.7% above an oil palm plantation (*E. guineensis* × *E. oleifera* hybrid) in Malaysia (Misztal et al., 2011), 0.16% and 0.24% above two *P. halepensis* (Aleppo pine) forests in Israel (Seko et al., 2017).

There was an order of magnitude difference between the proportion of net assimilated carbon released as VOCs between a Mediterranean oak-hornbeam forest (Schallhart et al., 2016) and a boreal evergreen forest. Also, a middle European poplar plantation (Brilli et al., 2015) clearly released a higher proportion of the assimilated carbon as VOCs than a boreal site in this study. The spruce forest in central Europe came closest to the results in Hyytiälä. These findings strongly imply that there are significant differences between the ecosystems in how they allocate carbon to VOCs, however the reasons for that are rather related to light and thermal conditions, species selection and age structure and soil properties than to the efficiency of an ecosystem to produce VOCs as an indefinable concept. In boreal ecosystems with relatively northern locations, the majority of carbon assimilation is concentrated within a couple of months around mid-summer with very short nights. The structure of forest and the tree species are effective in utilizing the high light availability during the summer months, whereas in more southern locations the light availability and thermal conditions allow more even carbon assimilation throughout a considerably longer period. This partly explains the lower proportion of C released as VOCs determined in this study, when compared to those measured in more southern locations. Additionally, the constitutive emission capacities of boreal evergreen species are known to be low when compared to deciduous species more common in central and southern Europe (Rinne et al., 2009; Ghirardo et al., 2010).

**3.3 Comparison between PTR-TOF and PTR-Quad measurements**

Generally, the EC method detected more masses than the SLP measurements, which can be explained by the preselection of the masses to be measured by the PTR-Quad and its low duty cycle. The comparison between PTR-TOF using the EC method and the PTR-Quad using the surface layer profile method (SLP, see Rantala et al., 2015) between April and June revealed 12 more compounds with exchange using the EC method (see Table 1, 2 and 3). The PTR-TOF measures all VOCs in a certain mass range, while the PTR-Quad needs a preselection of masses, limiting the number of recorded compounds. However, the classification if a compound has a measurable flux is different between the two methods (Sect. 2.3 and Rantala et al., 2015). Even though the EC method found almost twice the number of compounds with exchange, the total exchange was in on the same order of magnitude for both methods. The discrepancy of between the results of the two measurements are is mainly due to instrument and method differences. The horizontal distance between the inlets of these two instruments was just 25 m and the PTR-Quad was measuring from 13 m higher (calculated height 36 m) than the PTR-TOF, leading to a larger footprint area for the SLP method. This can be seen in Fig. 8, where the daytime (07:00 to 19:00) footprint calculated according to Kormann and Meixner, (2002) for the SLP and EC method are presented. Even though the nominal measurement height of the SLP was 36 m, the shown footprint is calculated for 33 m, as this was the closest height with horizontal wind measurements.

The main compounds (methanol, acetone and monoterpenes) were detected with by both methods. $C_4H_9^+$ (57.0699 Da amu) could not be measured by the PTR-Quad, as the unit mass resolution of the instrument is unable to separate the signal

from the water cluster isotope $H_7O_2{}^{18}O_1{}^+$ (57.0432 amu). On the other hand, the SLP measurements observed a flux of ethanol+formic acid which was not detected by the PTR-TOF. Interestingly, formic acid fluxes have also been observed at SMEAR II using the EC method with an Iodide-Adduct High-Resolution Time-of-Flight Chemical Ionization Mass Spectrometer in 2014 (Schobesberger et al., 2016). Other VOC fluxes, which were not detected by the PTR-TOF were methyl ethyl ketone (73 amu), a fragment of the green leave volatiles (83 amu) and MBO (87 amu). The formaldehyde fluxes were not included in the comparison, as the PTR-TOF detected them just during the first 8 days in June, during which the PTR-Quad had technical problems resulting in less than ten overlapping data points from the two instruments.

Overall, the flux values used for the comparison were small, as the compared data included values from April and May, and unfortunately the SLP measurements were not working during the warm, i.e. high flux, period in the beginning of June. Thus, the comparison was done using flux values that were mostly close to the detection limits of the EC and the SLP setups. The uncertainties of the turbulence measurements and noise of the VOC measurements, together with the different measurement setups, methods and footprints was seen in the scatter of the compared data, which affected correlation, fitting parameters and their uncertainties (Table 3).

However, the magnitude of the studied fluxes as well as their diurnal patterns were comparable for methanol, acetone, isoprene and monoterpenes (Table 3 and Fig. 9). The monoterpene and  methanol fluxes measured by the two methods  showed the highest  correlation, which was barely above 0.6  (Fig. 10). The fitted slopes of the scatter plots for acetone, isoprene and the monoterpenes were far from unity, as best $R^2$ values were calculated when using high intercepts.

[revised manuscript text omitted]

(*) Values under the limit of detection (2 $\sigma_{ind}$). $\sigma_{ind}$ was calculated using the propagation of error formula and the standard deviation at the borders of the individual 30 min CCFs.

**Table 2: Comparison between different studies using a PTR-TOF with the EC method. The listed compounds are limited to the ones measured in Hyytiälä. Numbers in parentheses describe deposition and emission, respectively. All values are 24 h averages, except**  **values marked with [b]. Emissions of butene & butanol are in red as they are anthropogenic (Sect. 3.4) and, therefore, not included in the total exchange.**

| | | net flux [nmol m$^{-2}$ s$^{-1}$] | | | | | |
|---|---|---|---|---|---|---|---|
| mass [Da] | elem. comp. | this study (21 days June) | Schallhart et al. (2016) | Park et al. (2013)[a] | Brilli et al. (2015) | this study (21 days June)[b] | Kaser et al. (2013a)[b] |
| 33.0335 | $CH_5O^+$ | 0.965* (-0.044/1.010) | 1.168 (-0.365/1.533) | 1.655 (-0.102/1.757) | 0.884 | 2.09[b] | 3.53[b] |
| 41.0386 | $C_3H_5^\pm$ | 0.050* (-0.001/0.051) | | 0.085 (-0.005/0.089) | | 0.10[b] | |
| 42.0338 | $C_2H_4N^+$ | 0.003 (-0.008/0.011) | 0.046 (-0.005/0.051) | | | <0.01[b] | |
| 43.0178 | $C_2H_3O^+$ | 0.027* (-0.011/0.038) | | 0.075 (-0.001/0.076) | | 0.07[b] | |
| 45.0335 | $C_2H_5O^+$ | 0.099* (-0.004/0.103) | 0.228 (-0.001/0.229) | 0.133 (-0.016/0.148) | 0.004 | 0.17[b] | 1.05[b] |
| 57.0699 | $C_4H_9^\pm$ | 0.199 (0/0.199) | | 0.016 (-0.011/0.027) | | 0.30[b] | |
| 59.0491 | $C_3H_7O^+$ | 0.297* (-0.001/0.297) | 0.335 (-0.01/0.345) | 0.281 (-0.004/0.286) | 0.035 | 0.48[b] | 0.13[b] |
| 60.0471 | unknown | 0.022* (-0.005/0.026) | | | | 0.03[b] | |
| 61.0284 | $C_2H_5O_2^\pm$ | 0.044* (-0.003/0.048) | 0.214 (-0.096/0.311) | 0.413 (-0.005/0.418) | | 0.09[b] | 1.64[b] |
| 67.0542 | $C_5H_7^\pm$ | 0.006* (-0.001/0.007) | | 0.012 (-0.004/0.017) | | 0.01[b] | |
| 69.0352 | unknown | 0.013* (-0.001/0.013) | | | | 0.03[b] | |
| 69.0699 | $C_5H_9^\pm$ | 0.083* (-0.003/0.086) | 6.466 (0/6.466) | 0.025 (-0.001/0.025) | 1.009 | 0.17[b] | 5.87[b] |
| 70.0696 | unknown | 0.007* (-0.001/0.008) | | | | 0.01[b] | |
| 89.0386 | $C_7H_5^\pm$ | 0.001 (-0.004/0.005) | | | | >-0.01[b] | |
| 93.0699 | $C_7H_9^\pm$ | 0.020* (-0.001/0.021) | | 0.058 (0/0.058) | | 0.04[b] | |
| 99.0769 | unknown | 0.001 (-0.004/0.004) | | | | 0.01[b] | |
| 137.1325 | $C_{10}H_{17}^\pm$ | 0.430 (0/0.430) | 0.219 (-0.001/0.219) | 0.290 (0/0.290)[2] | 0.005 | 0.63[b] | 0.71[b] |
| Net flux in study | | 2.07 | 9.78 | 4.43 | 1.99 | 3.93[b] | 15.07[b] |
| Length of data set [days] | | 21 | 21 | 33 | 129 | 21 | 31 |
| Highest emitted compound | | methanol | isoprene | methanol | isoprene | methanol | MBO & isoprene |
| # of compounds with flux | | 17 | 29 | 494 | 13 | 17 | 15[c] |

(*) Values for downward flux are under the respective limit of detection (2 $\sigma_{ind}$). $\sigma_{ind}$ was calculated using the propagation of error formula and the standard deviation at the borders of the individual 30 min CCFs. ([a]) Park et al. (2013) published the 24 h values of the identified masses, therefore no comparison of the unidentified compounds was possible. ([b]) 8 h average daytime values (10:00 - 16:00). ([c]) 8 of the 15 compounds were only recorded after a hail storm event.

~~(*) Values for downward flux are under the respective limit of detection (2 $\sigma_{ind}$). $\sigma_{ind}$ was calculated using the propagation of error formula and the standard deviation at the borders of the individual 30 min CCFs. ([a]) Park et al. (2013) published the 24h values of the identified masses, therefore no comparison of the unidentified compounds was possible. ([b]) Kaser et al. (2013a) published average daytime fluxes (10:00 - 18:00) and therefore their results cannot be directly compared with the 24h average values. ([c]) 8 of the 15 compounds were only recorded after a hail storm event.~~

Table 3: Statistics of the major compounds of SLP and the EC flux measurements.  The EC fluxes are in bold whereas the SLP fluxes are written in normal text. The fitting parameters describe the slope (upper value) and the intercept (lower value) of the linear model between the EC and the SLP fluxes. $N$ is the number of the data points for each compound. The numbers in parenthesis are lower and upper quartiles. The unit for the mean, median, intercept and quantile values is nmol m$^{-2}$ s$^{-1}$.

| nominal mass | $R^2$ | mean | median | 5% and 95% quant. | fitting | $N$ |
|---|---|---|---|---|---|---|
| 33 (methanol)* | 0.374 | 0.447 | 0.204 (-0.177, 1.161) | -1.602, 2.920 | 0.810 ± 0.220 | 92 |
| | | **0.553** | **0.730 (-0.931, 1.910)** | **-2.107, 3.296** | 0.171 ± 0.306 | |
| 42 (acetonitrile) | 0.003 | -0.009 | -0.003 (-0.026, 0.005) | -0.052, 0.026 | 0.296 ± 1.545 | 55 |
| | | **-0.017** | **-0.050 (-0.103, 0.072)** | **-0.209, 0.228** | -0.015 ± 0.040 | |
| 45 (acetaldehyde) | 0.033 | 0.019 | 0.024 (-0.106, 0.131) | -0.382, 0.481 | -0.195 ± 0.310 | 49 |
| | | **0.063** | **0.114 (-0.116, 0.245)** | **-0.406, 0.507** | 0.067 ± 0.082 | |
| 59 (acetone) | 0.051 | 0.176 | 0.090 (0.004, 0.262) | -0.156, 0.696 | 0.242 ± 0.191 | 119 |
| | | **0.173** | **0.210 (-0.046, 0.373)** | **-0.469, 0.677** | 0.131 ± 0.074 | |
| 61 (acetic acid)*,a | 0.064 | 0.336 | 0.244 (0.059, 0.504) | 0.003, 1.052 | 0.116 ± 0.130 | 49 |
| | | **0.025** | **0.058 (-0.029, 0.093)** | **-0.115, 0.195** | -0.014 ± 0.061 | |
| 69  (isoprene & MBO) | 0.127 | 0.082 | 0.033 (0.005, 0.091) | -0.009, 0.383 | 0.274 ± 0.160 | 82 |
| | | **0.035** | **0.035 (-0.013, 0.052)** | **-0.054, 0.104** | 0.013 ± 0.025 | |
| 93 (tol. & p-cym.) | 0.089 | 0.138 | 0.081 (0.024, 0.209) | -0.063, 0.519 | 0.088 ± 0.061 | 88 |
| | | **0.027** | **0.037 (-0.013, 0.052)** | **-0.054, 0.104** | 0.015 ± 0.014 | |
| 137 (monoterpenes) | 0.364 | 0.282 | 0.208 (0.084, 0.365) | 0.023, 0.754 | 0.503 ± 0.123 | 116 |
| | | **0.261** | **0.225 (0.135, 0.349)** | **0.019, 0.583** | 0.118 ± 0.051 | |

(*) sensitivity was derived from the instrumental transmission curve for the PTR-Quad. (a) the acetic acid sensitivity was estimated

[Figure]

**Figure 1: A) Normalized CCFs for monoterpene measurements in June, without time shift correction. The shift between the two computer clocks is clearly visible. B) CCFs after correcting for lag time shifts. C) The CCFs of monoterpenes after the final lag time correction. In the final (third) step, the smoothed maximum of the CCF function (see Fig. 2) was sought for each compound and 30 min data individually, in a ± 10 s window of the previous lag time (step 1 & 2).**

[Figure]

**Figure 2: For each compound which had a maximum above 3 $\sigma_{noise}$, the deviation of the CCF maxima from zero was used as the lag time correction.**

[Figure]

**Figure 3: Contribution of each compound to the 24 h average net flux of the respective month. Bars with thick outlines and \* above them correspond to negative fluxes, where the absolute value was taken before plotting them in the logarithmic scale. See Table 1 for corresponding compound names. Emissions of $C_4H_9^+$ are highlighted in a red as they are anthropogenic (Sect. 3.4) and, therefore, are not included in the net flux.**

[Figure]

**Figure 4: Diurnal pattern of the 9 compounds with the highest fluxes, the remaining compounds being summed up as 'other'. The panels show the fluxes for  snowmelt (top), start of growing season (mid) and summer (bottom). The number of data points per hour is dependent on the quality criteria filtering and whether it is in an hour when the automatic background was measured. (\*) The butene & butanol exchange is anthropogenic and thereby not emitted by the forest (Sect. 3.4).**

[Figure]

**Figure 5: Scatterplot of 30 min isoprene and monoterpene flux. The gray data points are values where the PAR was smaller than 200 µmol m⁻² s⁻¹. Isoprene fluxes are very low and especially during low PAR conditions they are heavily affected by noise and a mirroring effect (Langford et al., 2015).**

[Figure]

**Figure 6:** Diurnal variation of the nine most emitted compounds during the 21 days of measurement in June. The remaining compounds are summed up as 'other'. The high variation in the flux seen in Fig. 4 (diurnal plot of 9 days of measurements in June) is reduced, as meteorological events (e.g. rain) have less impact on the result. (*) The butene & butanol exchange is anthropogenic and thereby not emitted by the forest (Sect. 3.4).

[Figure]

**Figure 7: Average net flux of the major carbon emitters. Compounds whose elemental composition could not be identified were disregarded. See Table 1 for the corresponding compound names. The butene & butanol emissions were disregarded in this figure.**

[Figure]

**Figure 8: Footprint of the SLP and EC method. The higher nominal measurement height of the SLP fluxes increases the footprint drastically, when compared to the EC measurements. This could be a possible cause for the discrepancies in the results.**

[revised manuscript text omitted]